# RETHINKING AND RED-TEAMING PROTECTIVE PERTURBATION IN PERSONALIZED DIFFUSION MODELS

## ABSTRACT

Personalized diffusion models (PDMs) have become prominent for adapting pre-trained text-to-image models to generate images of specific subjects using minimal training data. However, PDMs are susceptible to minor adversarial perturbations, leading to significant degradation when fine-tuned on corrupted datasets. These vulnerabilities are exploited to create protective perturbations that prevent unauthorized image generation. Existing purification methods attempt to red-team the protective perturbation to break the protection but often over-purify images, resulting in information loss. In this work, we conduct an in-depth analysis of the fine-tuning process of PDMs through the lens of shortcut learning. We hypothesize and empirically demonstrate that adversarial perturbations induce a latent-space misalignment between images and their text prompts in the CLIP embedding space. This misalignment causes the model to erroneously associate noisy patterns with unique identifiers during fine-tuning, resulting in poor generalization. Based on these insights, we propose a systematic red-teaming framework that includes data purification and contrastive decoupling learning. We first employ off-the-shelf image restoration techniques to realign images with their original semantic meanings in latent space. Then, we introduce contrastive decoupling learning with noise tokens to decouple the learning of personalized concepts from spurious noise patterns. Our study not only uncovers fundamental shortcut learning vulnerabilities in PDMs but also provides a comprehensive evaluation framework for developing stronger protection. Our extensive evaluation demonstrates its superiority over existing purification methods and stronger robustness against adaptive perturbation.

## 1 INTRODUCTION

The rapid advancements in text-to-image diffusion models, such as DALL-E 2 (Ramesh et al., 2022), Stable Diffusion (Rombach et al., 2022), and MidJourney (mid), have revolutionized the field of image generation. These models can generate highly realistic and diverse images based on textual descriptions, enabling a wide range of applications in creative industries, entertainment, and beyond. However, the capability to fine-tune these models for personalized generation using a small set of reference images has raised concerns about their potential misuse, such as generating misleading or harmful content targeting individuals (Van Le et al., 2023; Salman et al., 2023) or threatening the livelihood of artists by mimicking unique artistic styles without compensation (Shan et al., 2023).

To address these issues, several protective perturbation methods have been proposed to protect user images from unauthorized personalized synthesis (Šarčević et al., 2024; Deng et al., 2024a; Wang et al., 2024a). These methods aim to proactively make images resistant to AI-based manipulation by crafting adversarial perturbations (Salman et al., 2023; Liang et al., 2023), applying subtle style-transfer cloaks (Shan et al., 2023), or crafting misleading perturbation that causes model's overfitting (Liu et al., 2024b). The model trained on perturbed data will generate images that are poor in quality, and thus, the unauthorized fine-tuning fails. Despite the protection effectiveness, different from the protective perturbation crafted for fixed and off-the-shelf diffusion models, where the protection against unauthorized editing (Liang et al., 2023) can be well explained by the adversarial vulnerability of neural networks (Ilyas et al., 2019), and the sharpness of the latent space of VAE (Kingma & Welling, 2013; Guo et al., 2023; Xue et al., 2023), *the underlying mechanism for how protective perturbation disturbs the fine-tuning of the personalized diffusion model has not been explored yet.*

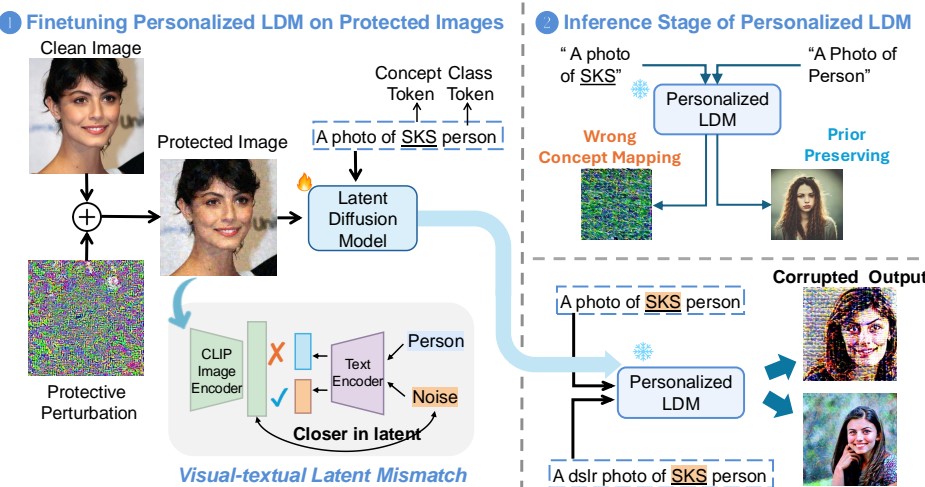

Figure 1: We observe that protective perturbation for personalized diffusion models creates a latent mismatch in the image-prompt pair. Fine-tuning on such perturbed data tricks the models, learning the wrong concept mapping. Thus, model generations suffer from severe degradation in quality.

Moreover, to systematically examine the practical performance of existing protection methods in the wild, purification studies (Cao et al., 2024; Zhao et al., 2024a) have been proposed with more advanced data purification process to further re-evaluate and red-teaming these protection methods. As demonstrated in Van Le et al. (2023), most of the protection methods lack resilience against simple purification like Gaussian smoothing. However, these traditional transformations also come with severe data quality degradation after purifying. Compared to these deterministic purifications, diffusion-based purification shows a stronger capacity to denoise the images and yield high-quality output by leveraging the distribution modeling ability of diffusion models. Based on the observation that clean images have better consistency upon reconstruction, IMPRESS (Cao et al., 2024) proposes optimization on the protected images to impose reconstruction consistency with visual LPIPS similarity constraints (Zhang et al., 2018). Despite effectiveness, IMPRESS is inefficient and requires a tremendous amount of time due to the iterative nature of the proposed optimization. On the other line, GrIDPure (Zheng et al., 2023) leverage pixel-space diffusion models to denoise the images by conducting an SDEdit process (Meng et al., 2021; Nie et al., 2022) that first converts the perturbed images into a slightly noisy state with a diffusion forward process and then denoise them back with a reverse process. To further improve visual consistency, GrIDPure divides the images into smaller grids with a small-step diffusion process. However, GrIDPure still yields *unfaithful* content that causes great change in identity due to the generative nature of the diffusion model. *How to design an effective, efficient, and faithful purification approach is still an open question.*

To gain better understanding, we first take a closer look at the fine-tuning process of PDMs through the lens of *causal analysis* and *shortcut learning* (Geirhos et al., 2020). We first build the underlying causal graph of learning on protected images, where we found protective perturbation manipulates the learning process by reinforcing the shortcut path from personalized identifier to injected noise. Furthermore, we found that existing effective protective perturbation introduces a latent-space misalignment between images and the textual prompts, where the perturbed images largely deviate from their original semantic concepts. This misalignment triggers the model to learn a shortcut connection between the identifier and more high-frequency and easy-to-learn noise patterns.

Based on these insights, we propose a systematic red-teaming framework motivated by causal intervention to empower robust PDMs against protective perturbations. Our approach conducts comprehensive purification from three perspectives, including input purification, contrastive decoupling learning and sampling. Compared to existing purification methods that are only limited to image purification, the advantages of our framework are three folds: i) *efficiency and faithfulness*: we conduct efficient one-shot image purification by using super-resolution and image restoration models that convert low-quality, noisy images into high-quality, purified ones; ii) *robustness and once-for-all*: we demonstrate that contrastive decoupling learning itself works alone and contributes

in robustness against adaptive perturbations crafted against our pipeline; iii) *system-level red-teaming*: not only limited to the input image, we propose systematic red-teaming strategies covering three stages including data purification, model training, and sampling strategy, offering a comprehensive evaluation on the effectiveness of future protection. We summarize our contributions as below:

- We uncover and empirically validate that protective perturbations work by exploiting the shortcut learning in PDMs with latent-space image-prompt misalignment from causal analysis.
- We propose a systematic red-teaming framework based on causal analysis that effectively mitigates these vulnerabilities through data purification and contrastive decoupled learning and sampling.
- We demonstrate the effectiveness, efficiency, and faithfulness of our approach through extensive experiments across 7 protections, showing significant improvements over existing methods. Our study provides a more systematic evaluation framework for future research on protective perturbations.

## 2 RELATED WORKS

**Data Poisoning as Protection against Unauthorized Training with LDMs.** Latent Diffusion Models (LDMs) (Rombach et al., 2022) have become dominant in various generative tasks, including text-to-image synthesis. To meet the demand for personalized generation, methods like Dream-Booth (Ruiz et al., 2023) have been proposed, which fine-tune LDMs using a small set of reference images to learn specific concepts. However, these advancements have raised concerns about potential misuse, such as generating misleading content targeting individuals (Van Le et al., 2023; Salman et al., 2023) and threatening the livelihood of professional artists through style mimicking (Shan et al., 2023). To address these issues, several data-poisoning-based methods have been proposed to protect user images from unauthorized personalized synthesis by injecting adversarial perturbations through minimizing adversarial target loss in image encoder or UNet denoiser (Salman et al., 2023), or denoising-loss maximization (Liang et al., 2023; Van Le et al., 2023; Liu et al., 2024b) or in opposite direction, denoising-loss minimization (Xue et al., 2023), or cross-attention loss maximization (Xu et al., 2024). Despite its effectiveness, the underlying mechanism of protection against diffusion model fine-tuning has not yet been explored well. To the best of our knowledge, Zhao et al. (2024a) is the only work that attempts to investigate the underlying mechanism. However, it is only limited to the vulnerability of the text encoder. *In this work, we provide a more comprehensive explanation from the view of latent mismatch and shortcut learning.*

**Data Purification that Further Breaks Protection.** Despite promising protection performance, studies (Van Le et al., 2023; An et al., 2024; Liu et al., 2024b) suggest that these perturbations without advanced transformation loss (Athalye et al., 2018) are brittle and can be easily removed under simple rule-based transformations. Among all types of transformation, state-of-the-art adversarial purification leverages diffusion models as purifiers to perturb images back to their clean distributions. In the classification scenario, DiffPure (Nie et al., 2022) is a mainstream approach for adversarial purification by applying SDEdit (Meng et al., 2021) on the poison with an off-the-shelf diffusion model. For purification against protective perturbation, GrIDPure (Lee & Chang, 2022) further adapts iterative DiffPure with small steps on multi-grid spitted image to preserve the original resolution and structure. However, due to their generative nature, these SDEdit-based purifications have limitations in yielding unfaithful content, where the purified images fail to preserve the original identity. Observing the perceptible inconsistency between the perturbed images and the diffusion-reconstructed ones, IMPRESS (Cao et al., 2024) conducts the purification via minimizing the consistency loss with constraints on the maximum LPIPS-based (Zhang et al., 2018) similarity change on pixel space. While it manages to preserve similarity, IMPRESS suffers from the inefficiency issue due to its iterative process and is ineffective under stronger protections like Liu et al. (2024b); Mi et al. (2024).

**Shortcut Learning and Causal Analysis.** Shortcut learning occurs when models exploit spurious correlations in training data, leading to poor generalization (Geirhos et al., 2020). The causal analysis provides a framework for addressing this by modeling cause-effect relationships (Pearl, 2009; Schölkopf et al., 2021). It helps identify true causal factors, distinguishing them from spurious correlations. In computer vision, models may incorrectly focus on background textures instead of object features (Brendel & Bethge, 2019). Techniques like Invariant Risk Minimization (Arjovsky et al., 2019) and Counterfactual Data Augmentation (Teney et al., 2021) leverage causal principles to improve robustness. In PDMs, protective perturbations can introduce spurious correlations between noise patterns and identifiers during fine-tuning. *Our work explores how to restore correct causal relationships when learning PDMs on perturbed data, which is under-explored in existing works.*

## 3 PRELIMINARY

**Personalized Latent Diffusion Models (LDMs) via DreamBooth Fine-tuning.** LDMs (Rombach et al., 2022) are generative models that perform diffusion processes in a lower-dimensional latent space, enhancing training and inference efficiency compared to pixel-space diffusion models (Ho et al., 2020). By conditioning on additional embeddings such as text prompts, LDMs can generate or edit images guided by these prompts. Specifically, an image encoder $\mathcal{E}$ maps an image $\mathbf{x}_0$ to a latent representation $\mathbf{z}_0 = \mathcal{E}(\mathbf{x}_0)$. A text encoder $\tau_\theta$ produces a text embedding $\boldsymbol{c} = \tau_\theta(c)$ for a given prompt $c$. The model trains a conditional noise estimator $\boldsymbol{\epsilon}_\theta$, typically a UNet (Ronneberger et al., 2015), to predict the Gaussian noise added at each timestep $t$, using the loss:

$$\mathcal{L}_{\text{denoise}}(\mathbf{x}_0, \boldsymbol{c}; \theta) = \mathbb{E}_{\mathbf{z}_0 \sim \mathcal{E}(\mathbf{x}_0), \boldsymbol{\epsilon}, t}\left[\|\boldsymbol{\epsilon} - \boldsymbol{\epsilon}_\theta(\mathbf{z}_0, t, \boldsymbol{c})\|_2^2\right]. \tag{1}$$

During inference, the model starts from random noise $\mathbf{z}_T \sim \mathcal{N}(0, \mathbf{I})$ and iteratively denoises it to obtain a latent $\tilde{\mathbf{z}}_0$, which is then decoded to generate the image $\tilde{\mathbf{x}}_0 = \mathcal{D}(\tilde{\mathbf{z}}_0)$. DreamBooth (Ruiz et al., 2023) fine-tunes a pre-trained LDM to generate images of specific concepts by introducing a unique identifier that links subject concepts and employing a class-specific prior-preserving loss to mitigate overfitting and language drift. The fine-tuning utilizes an instance dataset $\mathcal{D}_{\boldsymbol{x}_0} = \left\{\left(\boldsymbol{x}_0^i, \boldsymbol{c}^{\mathcal{V}^*}\right)\right\}_i$, and a class dataset $\mathcal{D}_{\bar{\boldsymbol{x}}_0} = \left\{\left(\bar{\boldsymbol{x}}_0^i, \bar{\boldsymbol{c}}\right)\right\}_i$, where $\boldsymbol{x}_0$ are subject images and $\bar{\boldsymbol{x}}_0$ are class images. The class-specific prompt $\bar{\boldsymbol{c}}$ is set as *"a photo of a [class noun]"*, and the instance prompt $\boldsymbol{c}^{\mathcal{V}^*}$ is *"a photo of $\mathcal{V}^*$ [class noun]"*, where $\mathcal{V}^*$ specifies the subject and "[class noun]" denotes the object category (e.g., "person"). The instance dataset contains the subject-specific images we want the model to learn, while the class dataset contains diverse images from the same category to prevent language drift. The fine-tuning process on these two datasets optimizes a weighted sum of the instance denoising loss and the prior-preservation loss:

$$\mathcal{L}_{db}(\mathbf{x}_0, \boldsymbol{c}^{\mathcal{V}^*}, \bar{\boldsymbol{x}}_0, \bar{\boldsymbol{c}}; \boldsymbol{\theta}) = \mathcal{L}_{\text{denoise}}\left(\mathbf{x}_0, \boldsymbol{c}^{\mathcal{V}^*}\right) + \lambda \mathcal{L}_{\text{denoise}}\left(\bar{\boldsymbol{x}}_0, \bar{\boldsymbol{c}}\right), \tag{2}$$

where $\lambda$ balances the two terms. With approximately 1k training steps and around four subject images, DreamBooth can generate vivid, personalized subject images (von Platen et al., 2022). **Protective Perturbation against Personalized LDMs.** Recent studies suggest that minor adversarial perturbation to clean images can significantly disturb the learning of customized diffusion and also prevent image editing with an off-the-shelf diffusion model by greatly degrading the quality of the generated image. Existing protective perturbation can be classified into two categories: perturbation crafted with fixed diffusion models and perturbation crafted with noise-model alternative updating. In this paper, we focus on the second category since they are more effective in the fine-tuning setting. The general framework of these protective perturbation methods is to craft noise that maximizes an adversarial loss $\mathcal{L}_{adv}$ that is typically designed as the denoising loss $\mathcal{L}_{\text{denoise}}$ and also alternatively update the noise generator surrogates $\theta'$ can be a single model (Van Le et al., 2023) or an ensemble of models (Liu et al., 2024b)) or the attention modules (Xu et al., 2024). Formally, at the $j$-th alternative step, the noise surrogate $\boldsymbol{\theta}'_j$ and perturbation $\boldsymbol{\delta}^{(j)}$ are updated via solving,

$$\boldsymbol{\theta}'_j \leftarrow \underset{\boldsymbol{\theta}'_{j-1}}{\arg\min} \sum_x \mathcal{L}_{db}\left(\mathbf{x} + \boldsymbol{\delta}^{(j-1)}, \boldsymbol{c}^{\mathcal{V}^*}, \bar{\boldsymbol{x}}, \bar{\boldsymbol{c}}; \boldsymbol{\theta}'_{j-1}\right); \boldsymbol{\delta}^{(j)} \leftarrow \underset{\left\|\boldsymbol{\delta}^{(j-1)}\right\|_\infty \leq r}{\arg\max} \mathcal{L}_{adv}\left(\mathbf{x} + \boldsymbol{\delta}^{(j-1)}, \bar{\boldsymbol{c}}; \boldsymbol{\theta}'_j\right). \tag{3}$$

To solve this, standard Gradient Descent is performed on the model parameter while the images are updated via Project Gradient Descent (PGD) (Madry et al., 2018) to satisfy the $\ell_\infty$-ball perturbation budget constrain with radius $r$,

$$\boldsymbol{\theta}_i \leftarrow \boldsymbol{\theta}_{i-1} - \beta \nabla_{\boldsymbol{\theta}_{i-1}} \mathcal{L}_{db}; \quad \mathbf{x}^{k+1} \leftarrow \Pi_{B_\infty(\mathbf{x}^0, r)}\left[\mathbf{x}^k + \eta \cdot \text{sign} \nabla_{\mathbf{x}^k} \mathcal{L}_{adv}\left(\mathbf{x}^k\right)\right], \tag{4}$$

where $\Pi_{B_\infty(\mathbf{x}^0, r)}(\cdot)$ is a projection operator on the $\ell_\infty$ ball that ensures $\mathbf{x}^k \in B_p(\mathbf{x}^0, r) = \left\{\mathbf{x}' : \|\mathbf{x}' - \mathbf{x}^0\|_\infty \leq r\right\}$, $\eta$ denotes the PGD step size and the total PGD step is $K$.

**Causal Analysis and Structural Causal Model.** Causal analysis models cause-and-effect relationships between variables (Pearl, 2009), helping identify spurious correlations and mitigate shortcut learning Geirhos et al. (2020). A Structural Causal Model (SCM) uses structural equations and a directed acyclic graph to represent causal relationships. It comprises endogenous variables $\mathbf{V}$, exogenous variables $\mathbf{U}$, and structural equations $f_i$, where each $V_i \in \mathbf{V}$ is defined as $V_i = f_i(\text{Pa}(V_i), U_i)$. By intervening on spurious correlations, causal analysis helps models focus on true causal relationships rather than superficial patterns. For more details, see Pearl (2009); Geirhos et al. (2020).

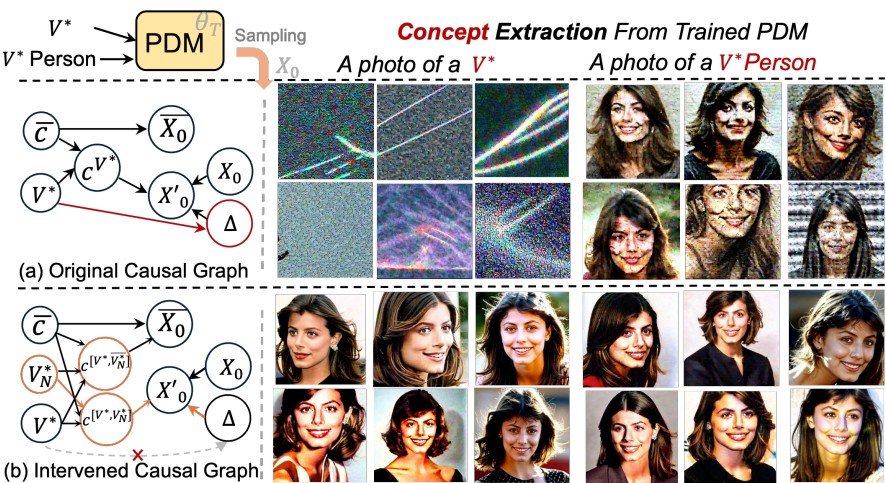

Figure 2: (a) The original causal graph representing the variable relationships in personalized diffusion model learning. Red arrows indicate the shortcut path introduced by protective perturbation. (b) Intervened causal graph with our proposed CDL. Orange arrows indicate our imposed path for decoupling noise after the intervention. With concept extraction, we examine that CDL alone helps the model learn the right correlations for linking identifier $\mathcal{V}^*$ and personalized concept $X_0$.

## 4 METHODOLOGY

### 4.1 PROTECTIVE PERTURBATION CAUSES LATENT-SPACE IMAGE-PROMPT MISMATCH

We first derive the formulation of learning personalized diffusion models on perturbed data. For the case of data poisoning, the instance data is perturbed by some adversarial noise $\boldsymbol{\delta}$, and the personalized diffusion models optimize the following loss,

$$\mathcal{L}_{db}^{adv}(\mathbf{x}_0, \boldsymbol{c}^{\mathcal{V}^*}, \bar{\boldsymbol{x}}_0, \bar{\boldsymbol{c}}; \boldsymbol{\theta}) = \mathcal{L}_{\text{denoise}}\left(\mathbf{x}_0 + \boldsymbol{\delta}, \boldsymbol{c}^{\mathcal{V}^*}\right) + \lambda \mathcal{L}_{\text{denoise}}\left(\bar{\boldsymbol{x}}_0, \bar{\boldsymbol{c}}\right). \tag{5}$$

Based on the adversarial loss in Eq. 5, with annotation of $\boldsymbol{c}^{\mathcal{V}^*} = \bar{\boldsymbol{c}} \oplus \mathcal{V}^*$ where $\mathcal{V}^*$ denotes the embedding of the unique identifier, we build the underlying causal graph (Pearl, 2009) in (a) of Fig. 2 (See App. C.1 on the construction details) to represent the learning process of the personalized diffusion model for linking personalized identifier to instance concept. We use the upper letter to represent random variables and the lower letter to represent the value instance. From this graph, we found that there is an unintended association (colored in red) derived from the instance condition $\mathcal{V}^*$ to the injected noise variable $\Delta$. In an ideal scenario, the protective perturbation represents a completely relevant concept and should be independent of both the class-prior prompt $\bar{\boldsymbol{c}}$ and the unique identifier $\mathcal{V}^*$. However, during training, the model observes pairs of perturbed images $\mathbf{x}_0 + \boldsymbol{\delta}$ and instance prompts $\mathbf{c}^{\mathcal{V}^*}$, leading to unintended associations between $\Delta$ and $\mathcal{V}^*$ in the causal graph. To validate this, we prompt the model trained on perturbed data to generate images on two different prompts, *"a photo of $\mathcal{V}^*$"* and *"a photo of $\mathcal{V}^*$ Person"*. As we can see from Fig. 2, the model erroneously attributes the noise patterns to $\mathcal{V}^*$ and thus generates noisy portraits for "$\mathcal{V}^*$ Person".

We defined the path $\mathcal{V}^* \to \Delta$ as identifier-noise shortcut for the following analysis. To establish and reinforce this shortcut path, we found that one important property that effective perturbation methods should have is the ability to cause *latent-space image-prompt mismatch*. That is, the images and their corresponding prompts are not semantically aligned in the latent space after the perturbation. Then thus, when learning on such pairs, it will create contradiction and force the models to dump that chaotic perturbation pattern into the rarely-appeared identifier token $\mathcal{V}^*$ instead of learning the clean identity behind $\boldsymbol{x}_0$. We infer it based on two empirical observations: i) random perturbation with the same strength does not affect the learning performance of the personalized diffusion model; ii) the generated portraits using the perturbed diffusion model usually have lower quality and larger image distortion than the slightly perturbed input images. The first observation justifies that if the

perturbation does not cause a significant latent shift, then the learning of the personalized diffusion model will not be affected, while the second observation suggests that the perturbed model learns more abstract noise concepts instead of just the noise pattern in the input pixel space. We further validate this through the following experiments of latent-mismatch visualization and concept interpretation.

Specifically, using the paired CLIP encoders, we first embed the latent of clean and perturbed images and also embed the textual concept of a person with a list of prompts describing the person concept, such as *"a photo of person's face"*. Then, we leverage three distinct 2D visualization techniques, including TSNE (Maaten & Hinton, 2008), Truncated-SVD (Halko et al., 2011), and UMAP (McInnes et al., 2018) on image-prompt embedding pairs. The results in Fig. 3 suggest that protective perturbation indeed significantly shifts the portrait latent from its original region of the "person" concept. Moreover, we precisely split the latent space into two regions with a zero-

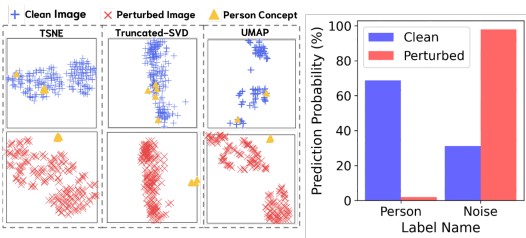

Figure 3: Latent 2D visualization and concept classification of images using CLIP encoders.

shot CLIP-based classifier, where we find that the perturbed images have a higher probability of being classified into the "noise" region instead of the "person" region in latent space. Please refer to Fig. 7 and Fig. 9 in the App. B.2 for more interpretation and visualization experiments.

These findings indicate protective perturbation indeed leads to latent mismatch. This latent mismatch creates an opportunity for shortcut learning (Geirhos et al., 2020; Hermann et al., 2023), where models optimize for easily accessible features rather than robust predictive patterns. In our case, PDMs face a binary choice: linking the unique identifier $\mathcal{V}^*$ either to the noise $\Delta$ or to the person identity concept $X_0$. From the perspective of loss minimization efficiency, PDMs naturally gravitate toward learning the high-frequency noise patterns rather than the more complex and desired person identity concept $X_0$, as this provides a computationally easier path to reduce training loss.

### 4.2 TRAINING CLEAN PDMs ON PERTURBED DATA WITH SYSTEMATIC RED-TEAMING

To address this shortcut learning issue, we propose a systematic red-teaming framework inspired by causal intervention (Geirhos et al., 2020), which is a widely used technique to mitigate shortcut learning in traditional machine learning tasks. Causal intervention (Kaddour et al., 2022) usually involves data argumentation or modifying the training process to disrupt spurious correlations. To mitigate shortcut learning in PDMs, we propose two key strategies: (i) *Removing Noise Variables* through image restoration techniques to eliminate adversarial noise and realign images with their true semantic representations; and (ii) *Weakening Spurious Paths and Strengthening Causal Paths* via Contrastive Decoupling Learning, which disentangles personalized concepts from noise patterns by incorporating noise tokens into prompts and leveraging clean prior data. We detail these approaches below and summarize our framework in Algorithm 1. Please refer to the App. C.2 for more details.

**Image Purification via Image Restoration.** An intuitive and effective approach to removing the direct influence of adversarial noise is to purify the input images using image restoration techniques. We view the perturbed images as degraded images in the image restoration domain (Wang et al., 2021) and leverage off-the-shelf image restoration models to convert low-quality, noisy images into high-quality, purified ones. Specifically, we use a face-oriented model named CodeFormer (Liu et al., 2023), which is trained on facial data to restore images based on latent code discretization. To further enhance the purification of non-face regions, we employ an additional diffusion-based super-resolution (SR) model. Compared to previous state-of-the-art optimization-based purification methods (Cao et al., 2024) and diffusion-based purification methods (Zhao et al., 2024a), this simple yet effective pipeline yields faithful purified images with better efficiency since it only requires a single inference pass. We term this module *CodeSR* as it combines CodeFormer and SR in sequence.

**Contrastive Decoupling Learning (CDL).** To further mitigate shortcut learning, we introduce Contrastive Decoupling Learning, which aims to disentangle the learning of desired personalized concepts from undesired noise patterns. We achieve this by augmenting the prompts with additional tokens related to the noise pattern, denoted as $\mathcal{V}_N^*$, such as *"XX noisy pattern"*. Ideally, these newly

---

**Algorithm 1** Training Clean Personalized LDMs on Perturbed Data with Systematic Red-Teaming

---

**Input:** Corrupted training set $X_0'$, pre-trained LDM $\theta_0$, CodeFormer $\phi = \{\mathcal{E}_\phi, \mathcal{D}_\phi, \mathcal{T}_\phi, \mathcal{C}\}$, SR model $\psi$, prior data $\bar{X}_0$, noise token $\mathcal{V}_N^*$, personalized identifier $\mathcal{V}^*$, instance prompt $c^{\mathcal{V}^*}$, class prompt $c$, number of generations $N_{\text{gen}}$

**Output:** Personalized diffusion model with clean-level generation performance $\theta_T$

1: **Step 1: Input Purification with CodeFormer and Super-resolution Model**
2: *CodeFormer*: Predict code $\tilde{Z}_c = \mathcal{T}_\phi(\mathcal{E}_\phi(X_0'), \mathcal{C})$; obtain high-quality restoration $\tilde{X}_0 = \mathcal{D}_\phi(\tilde{Z}_c)$
3: *Super-resolution*: Resize $\tilde{X}_0$ to $128 \times 128$; apply SR model $\psi$ to obtain $\tilde{X}_0^{\text{purified}}$ at $512 \times 512$
4: **Step 2: Contrastive Decoupling Learning**
5: **for** $i = 1$ **to** $T$ training steps **do**
6:     Sample instance data $x_i$ from $\tilde{X}_0^{\text{purified}}$, and class-prior data $\bar{x}_0$ from $\bar{X}_0$
7:     Craft decoupled instance prompt $c_{\text{dec}}^{\mathcal{V}^*} = \texttt{concat}(c^{\mathcal{V}^*}, \mathcal{V}_N^*)$ and class-prior prompt $c_{\text{dec}} = \texttt{concat}(c, \text{``without''}, \mathcal{V}_N^*)$
8:     Optimize the LDM $\theta_i$ with standard DreamBooth loss $\mathcal{L}_{\text{db}}$     ▷ Following Eq. 2
9:     $\mathcal{L}_{\text{db}}(x_i, c_{\text{dec}}^{\mathcal{V}^*}, \bar{x}_0, c_{\text{dec}}; \theta_i) = \mathcal{L}_{\text{denoise}}(x_i, c_{\text{dec}}^{\mathcal{V}^*}) + \lambda \mathcal{L}_{\text{denoise}}(\bar{x}_0, c_{\text{dec}})$
10:     Update LDM $\theta_i$ with $\nabla_{\theta_i} \mathcal{L}_{\text{db}}$ using AdamW optimizer on UNet Denoiser and Text Encoder
11: **end for**
12: **Inference:** Perform decoupled sampling $\{X_{\text{gen}}^j\}_{j=1}^{N_{\text{gen}}}$ with the trained PDM   ▷ Following Eq. 6

---

added tokens absorb all the noise components in the image, leaving the clean, personalized concept associated with the personalized identifier $\mathcal{V}^*$. During training, we insert $\mathcal{V}_N^*$ into the prompt of instance data with the suffix *"with XX noisy pattern"*, and include the "inverse" of $\mathcal{V}_N^*$ in the prompt of class-prior data with the suffix *"without XX noisy pattern"*. This contrastive prompt design encourages the model to distinguish between the instance concept and noise patterns, thus weakening spurious correlations. During inference, we add the suffix *"without XX noisy pattern"* to the prompt input to guide the model in disregarding the learned patterns associated with $\mathcal{V}_N^*$, thereby generating images that focus on the personalized concept. Furthermore, by using classifier-free guidance (Ho & Salimans, 2022) with a negative prompt $c_{\text{neg}} = $ *"noisy, abstract, pattern, low quality"*, we can further guide the trained model to generate high-quality images related to the learned concept. Specifically, given timestamp $t$, we perform sampling using the linear combination of the good-quality and bad-quality conditional noise estimates with guidance weight $w^{\text{neg}} = 7.5$:

$$\tilde{\epsilon}_\theta(\mathbf{z}_t, \mathbf{c}) = (1 + w^{\text{neg}})\epsilon_\theta\left(\mathbf{z}_t, \mathbf{c}^{[\mathcal{V}^*, \bar{\mathcal{V}}_N^*]}\right) - w^{\text{neg}}\epsilon_\theta(\mathbf{z}_t, \tau_\theta(c_{\text{neg}})) \tag{6}$$

## 5 EXPERIMENTS

### 5.1 EXPERIMENTAL SETUP

**Datasets and Metrics.** Our experiments are mainly performed on the VGGFace2 (Cao et al., 2018) face dataset following (Van Le et al., 2023; Liu et al., 2024b). Four identities are selected from each dataset, and we randomly pick eight images from each individual and split those images into two subsets for image protection and reference. Moreover, we also visually demonstrate the purification ability of our approach on samples from an artwork painting dataset, WikiArt (Saleh & Elgammal, 2015), and the CelebA (Liu et al., 2015). For the metric, we evaluate the generated images in terms of their *semantic-related quality* and *graphical aesthetic quality*. For the semantic-related score, we compute the cosine similarity between the embedding of generated images and reference images, which we term the Identity Matching Similarity (IMS) score. We reported the weighted averaged IMS score by employing two face embedding extractors, including *antelopev2* model from InsightFace library (Deng et al., 2020) following IP-adapter (Ye et al., 2023) and *VGG-Net* (Simonyan & Zisserman, 2014) from Deepface library (Taigman et al., 2014) following (Van Le et al., 2023). The IMS score is computed via a weighted sum: IMS$= \lambda$IMS$_{\text{IP}} + (1 - \lambda)$IMS$_{\text{VGG}}$, where $\lambda$ is set as 0.7. For the graphical quality $Q$, we report the average of two metrics: i) *LIQE* (Zhang et al., 2023a) (with re-normalization to $[-1, +1]$); ii) *CLIP-IQAC* following (Liu et al., 2024b), which is based on CLIP-IQA (Wang et al., 2023a) with class label. See App. A.1 for details.

**Purification Baselines and Perturbation Methods.** For *purification baselines*, we consider both model-free and diffusion-based approaches. The model-free methods include ❶ Gaussian Filtering,

Table 1: Results of different purification methods under different protective perturbations. The best performances are in **bold**, and second runners are shaded in gray. ∗ denotes significant improvement that passes the Wilcoxon signed-rank significance test with $p \leq 0.01$.

| Methods | Clean | | FSMG | | ASPL | | EASPL | | MetaCloak | | AdvDM | | PhotoGuard | | Glaze | |
|---|---|---|---|---|---|---|---|---|---|---|---|---|---|---|---|---|
| | IMS↑ | Q↑ | IMS↑ | Q↑ | IMS↑ | Q↑ | IMS↑ | Q↑ | IMS↑ | Q↑ | IMS↑ | Q↑ | IMS↑ | Q↑ | IMS↑ | Q↑ |
| Clean | -0.13 | 0.15 | -0.13 | 0.15 | -0.13 | 0.15 | -0.13 | 0.15 | -0.13 | 0.15 | -0.13 | 0.15 | -0.13 | 0.15 | -0.13 | 0.15 |
| Perturbed | - | - | -0.43 | -0.54 | -0.67 | -0.52 | -0.62 | -0.50 | -0.35 | -0.53 | -0.27 | -0.36 | -0.18 | -0.24 | -0.28 | -0.28 |
| Gaussian F. | -0.23 | -0.52 | -0.19 | -0.55 | -0.20 | -0.57 | -0.17 | -0.58 | -0.07 | -0.63 | -0.11 | -0.57 | -0.23 | -0.53 | -0.18 | -0.54 |
| JPEG | -0.27 | -0.13 | -0.15 | -0.41 | -0.21 | -0.52 | -0.27 | -0.50 | -0.34 | -0.38 | -0.15 | -0.02 | -0.13 | 0.07 | -0.19 | -0.03 |
| TVM | -0.15 | -0.64 | -0.12 | -0.65 | -0.16 | -0.66 | -0.10 | -0.67 | -0.11 | -0.69 | -0.12 | -0.65 | -0.15 | -0.64 | -0.11 | -0.66 |
| PixelDiffPure | -0.34 | -0.60 | -0.41 | -0.57 | -0.43 | -0.54 | -0.57 | -0.61 | -0.28 | -0.58 | -0.40 | -0.55 | -0.25 | -0.55 | -0.41 | -0.59 |
| L.DiffPure-∅ | -0.24 | 0.16 | -0.07 | -0.47 | -0.36 | -0.59 | -0.22 | -0.49 | -0.52 | -0.43 | -0.55 | -0.24 | -0.12 | -0.40 | -0.38 | -0.42 |
| L.DiffPure | -0.28 | 0.21 | -0.25 | -0.45 | -0.31 | -0.61 | -0.30 | -0.46 | -0.31 | -0.51 | -0.57 | -0.30 | -0.25 | -0.47 | -0.41 | -0.47 |
| DDSPure | -0.25 | -0.38 | -0.15 | -0.34 | -0.05 | -0.38 | -0.08 | -0.39 | -0.16 | -0.49 | -0.19 | -0.43 | -0.12 | -0.37 | -0.22 | -0.41 |
| GrIDPure | -0.46 | -0.17 | -0.10 | -0.20 | -0.21 | -0.16 | -0.13 | -0.25 | -0.23 | -0.25 | -0.09 | -0.25 | -0.03 | -0.22 | -0.24 | -0.13 |
| IMPRESS | -0.02 | -0.18 | -0.15 | -0.53 | -0.16 | -0.49 | -0.29 | -0.64 | -0.34 | -0.29 | -0.34 | -0.34 | -0.16 | -0.21 | -0.10 | -0.43 |
| Ours | 0.14* | 0.54* | 0.23* | 0.65* | 0.09 | 0.62* | 0.09* | 0.63* | 0.38* | 0.58* | 0.29* | 0.67* | 0.24* | 0.63* | 0.31* | 0.66* |

which reduces noise and detail using a Gaussian kernel; ❷ Total Variation Minimization (TVM), which reconstructs images by minimizing the difference between original and reconstructed images while enforcing smoothness; and ❸ JPEG Compression, which reduces image file size by transforming images into a compressed format. The diffusion-based methods include ❹ (Pixel)DiffPure (Nie et al., 2022), which leverages pretrained pixel-space diffusion models to smooth adversarial noise with small-step SDEdit process (Meng et al., 2021); ❺ LatentDiffPure, which is developed in the paper similar as DiffPure but with LDM as a purifier (two variants w/ and w/o prompting); ❻ DDSPure (Carlini et al., 2022), which finds an optimal timestamp for adversarial purification with SDEdit process; ❼ GrIDPure (Zheng et al., 2023), which further conducts iterative DiffPure with small steps with grid-based splitting to improve structure similarity; and ❽ IMPRESS (Cao et al., 2024), which purifies by optimizing latent consistency with visual similarity constraints. *For protective perturbation*, we consider six of existing SoTA approaches, including perturbation crafted with bi-level optimization, such as *FSMG, ASPL, EASPL* (Van Le et al., 2023), *MetaCloak* (Liu et al., 2024b), and perturbations crafted with adversarial perturbation with fixed models, such as *AdvDM* (Liang et al., 2023), *PhotoGuard* (Salman et al., 2023), and *Glaze* (Shan et al., 2023). For each setting, we set the perturbation to be ASPL by default. We set the $\ell_\infty$ radius to 11/255 with a six-step PGD step size of 1/255 by default following (Van Le et al., 2023). See App. A.2 for more details.

## 5.2 Effectiveness, Efficiency, and Faithfulness

**Effectiveness Comparison.** We present the effectiveness of different purification across seven perturbation methods in Tab. 1. From the table, we can see that compared to the clean case, training on perturbing data causes serve model degradation from both identity similarity and image quality. Across all perturbations, ASPL causes the most severe degradation under the setting without purification, while MetaCloak performs more robustly under rule-based purification. Compared to rule-based purification, diffusion-based approaches achieve better performance in improving both identity similarity and image quality in most settings. Among them, GrIDPure yields relatively better purification performance since it considers the structure consistency, which suppresses the generative nature during the purification. However, there are still gaps in the IMS score compared to the clean case, and most of the quality scores after conducting GrIDPure purification are still negative. Compared to these baselines, our method closes the gap by further improving the IMS and quality scores, which are even higher than the clean training case in all the settings. The reasons are twofold: first, we use image-restoration-based approaches, which preserve the image structure well; furthermore, our CDL module contributes significantly to quality improvement. Please refer to the App. B for the full comparison results with standard deviations.

**Efficiency and Faithfulness of Purification.** We present the evaluation of time cost and purification faithfulness compared to all other diffusion-based purification approaches in Tab. 2. The time cost is measured in seconds per sample with consideration of model loading. Compared to other methods, our purification has the lowest time cost and is 10× faster than the previous SoTA method, IMPRESS. The reason behind this is that we leverage the super-resolution module, which empowers the usage of skip-step sampling to boost the generation time. Moreover, we test the purification faithfulness of each method in terms of LPIPS loss (Zhang et al., 2018), a common metric measuring the visual

Figure 4: Visualization of purified images that were originally protected by MetaCloak. Our method shows high faithfulness and high quality, while others fail to effectively purify the perturbation.

perception distance of two images. From Tab. 2, we can see that our method achieves the lowest LPIPS loss. To visually validate this, we additionally present the purified images in Fig. 4. From the figure, we can see that other diffusion-based approaches have limitations in hallucinating the content, introducing severe artifacts, or not having enough purification strength. In particular, we observed that LatentDiffPure causes a great change in identity during the purification, which might be attributed to the semantic distortion during the purification process in latent space. On the other hand, GrIDPure (Zhao et al., 2024a) brings some artifacts to the purified image, which indicates that the underlying unconditional diffusion model pre-trained on ImageNet might not be suitable for general domain purification. In comparison, our purification method significantly enhances faithfulness by leveraging off-the-shelf image restoration models. These models are designed to preserve the structural integrity of the input, resulting in output images that closely maintain the original composition while effectively removing perturbations. This approach ensures that the purified images retain the essential features and identity of the original subjects, while successfully mitigating unwanted artifacts or noise.

Table 2: Faithfulness and efficiency of different diffusion-based purifications.

| Methods | LPIPS ↓ | Time Cost↓(s) |
|---|---|---|
| IMPRESS | 0.451 | 675 |
| PixelDiffPure | 0.495 | 102 |
| DDSPure | 0.384 | 122.5 |
| GrIDPure | 0.429 | 92.75 |
| LatentDiffPure | 0.453 | 63.75 |
| LatentDiffPure-∅ | 0.450 | 63.25 |
| **Ours** | **0.271** | **51** |

Table 3: Effectiveness of different model variants against Adaptive Attacks (AA).

| Modules | CDL | Before AA | | | After AA | | | $\mathbb{E}$[Avg.] |
|---|---|---|---|---|---|---|---|---|
| | | IMS | Q | Avg. | IMS | Q | Avg. | |
| CodeSR | ✓ | 0.256 | **0.514** | **0.385** | 0.116 | **-0.070** | 0.023 | **0.204** |
| | ✗ | -0.215 | 0.028 | -0.094 | -0.313 | -0.533 | -0.423 | -0.259 |
| Code | ✓ | **0.294** | 0.385 | 0.339 | 0.138 | -0.104 | 0.017 | 0.178 |
| | ✗ | -0.336 | 0.020 | -0.158 | -0.382 | -0.474 | -0.428 | -0.293 |
| SR | ✓ | 0.190 | 0.260 | 0.225 | **0.249** | -0.182 | **0.034** | 0.130 |
| | ✗ | -0.059 | -0.439 | -0.249 | -0.114 | -0.616 | -0.365 | -0.307 |

## 5.3 RESILIENCE AGAINST ADAPTIVE PERTURBATIONS

DNN-based purification is prone to further adaptive attacks due to the non-smoothness in terms of latent representation space (Guo et al., 2023) and also the vulnerability by exploiting adversarial examples (Ilyas et al., 2019). To validate whether our framework can still work upon adaptive adversarial perturbation with new knowledge of our pipeline, we additionally conduct experiments on evaluations of different variants of our approach before and after the adaptive perturbation crafted against the image purification part. The adversarial perturbation is crafted following AdvDM with consideration of the CFG (Ho & Salimans, 2022) sampling trajectory with a large perturbation budget of $r = 16/255$. For the model variants, we consider the full variant with both modules turned on, as well as the ablated versions with one of them turned off. From Tab. 3, we can see that the full variant with CDL is robust to the adaptive attack across other variants in terms of performance drop.

Furthermore, we notice that the variant with both SR and CDL yields a slightly better average score than the CodeSR configuration after the attack. This indicates that the CodeFormer module might be more susceptible to the adaptive attack while the SR module is more robust. However, using SR with CDL solely in case cases gives sub-optimal purification results. Our CodeSR configuration with CDL gives a better expected overall performance under mixed perturbation scenarios with $P(\text{AA})=50\%$.

## 5.4 ABLATION STUDY AND SENSITIVITY ANALYSIS

**Contribution of Individual Modules.** We present ablations on the three modules in our method in Tab. 4. From the table, our method works best under the full setting. When turning off any of the modules, the average performance degrades, with turning off CDL suffers the most. On the other hand, if we only turn on one of the modules, we find that CDL is still the most important one that retains higher generation performance. Furthermore, if we only do input purification without CDL, the generation quality is not as good as the full setting with CDL. This indicates that CDL is crucial for the performance of our method. Surprisingly, when only enabling the SR module, the IMS score is relatively good but with bad quality. While turning on the CodeFormer module alone, the boost is more on the quality score side. The settings that enable these two modules together yield a higher average score. These indicate that SR and CodeFormer modules are complementary to each other. Furthermore, for the settings that only allow two modules enabled, we found that the combination of CodeFormer and CDL yields the best performance compared to the other two combinations. Furthermore, we visualize the quality-score curve of identifier $\mathcal{V}*$ that shows consistent improvement during training in the App. B.1. In conclusion, the results suggest that all modules contribute to the learning performance gain in both IMS and quality scores.

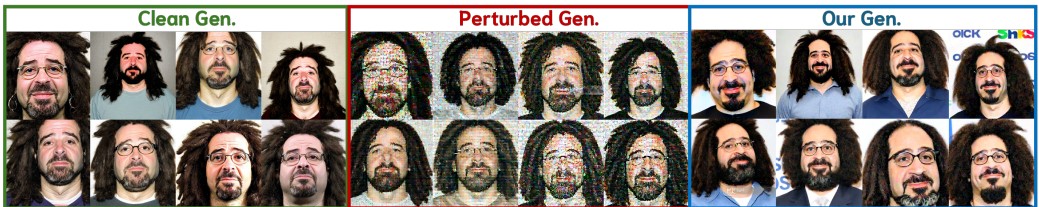

Figure 5: Generations from models trained on: (left) clean data, (middle) perturbed data without defense, and (right) purified data using our defense approach. The results demonstrate that our defense method significantly enhances generation quality, bringing it closer to clean data levels.

**Generation Visualization and Sensitivity Test.** We further visualize the generation of models trained in three cases, including clean, perturbed, and purified in Fig. 5. The visualization demonstrates that our defense greatly helps retain clean-level generation quality. Additionally, we find that the concept learned associated with $\mathcal{V}*$ under perturbed case matches the noise concept learned using CDL alone, indicating the CDL successfully decouples the learning of noise patterns (refer to the App. B.2). The sensitivity analysis of noise tokens is provided in App. B.3.

Table 4: Ablation study on individual modules.

| Settings | | | Metrics | | |
|---|---|---|---|---|---|
| CodeF. | SR | CDL | IMS↑ | Q↑ | Avg. ↑ |
| ✓ | ✓ | ✓ | 0.256 | **0.514** | **0.385** |
| ✓ | ✓ | ✗ | -0.215 | 0.028 | -0.094 |
| ✓ | ✗ | ✓ | **0.294** | 0.385 | 0.339 |
| ✗ | ✓ | ✓ | 0.190 | 0.260 | 0.225 |
| ✓ | ✗ | ✗ | -0.336 | 0.020 | -0.158 |
| ✗ | ✓ | ✗ | -0.059 | -0.439 | -0.249 |
| ✗ | ✗ | ✓ | 0.160 | 0.038 | 0.099 |
| ✗ | ✗ | ✗ | -0.271 | -0.425 | -0.348 |

## 6 CONCLUSION

In this paper, we dive into the underlying mechanism behind the effectiveness of existing protective perturbation approaches against the unauthorized fine-tuning of personalized diffusion models. Motivated by the latent mismatch observation, we propose to use super-resolution and image restoration models for latent realignment. Furthermore, we propose contrastive decoupling learning with quality-enhanced sampling based on the analysis from the shortcut learning perspective. Extensive experiments demonstrate the effectiveness, efficiency, and faithfulness of our method. Despite being mainly tested on facial data, our framework can generalize to other domains beyond the facial domain. Future work could optimize module combinations for balanced utility and robustness (in Sec. 5.3), and develop stronger protection methods based on our framework's robustness-effectiveness trade-off.

# 7 REPRODUCIBILITY STATEMENT

To facilitate replication and further exploration of our work, we have made concerted efforts to provide comprehensive details about our methodologies. All code used for data preprocessing, model training, and evaluation is provided in the supplementary materials. The code is organized and documented to allow researchers to reproduce our results seamlessly. Instructions for setting up the computational environment, including software versions and dependencies, are included to ensure that others can replicate our setup accurately.

We utilized publicly available datasets such as VGGFace2, WikiArt, and CelebA. Detailed information on how to access these datasets and any preprocessing steps are provided in supplementary files. By using standard datasets, we aim to facilitate comparisons and validations by other researchers. Hyperparameters, model architectures, and training protocols are thoroughly described in Sec. 3 and 5, and further elaborated in App. A.2. We specify the number of training epochs, batch sizes, learning rates, and optimization algorithms used. Such detailed descriptions are intended to ensure that others can replicate our training process and verify our findings.

The metrics used for evaluation, including Identity Matching Similarity (IMS) and graphical quality (Q), are clearly defined in Section 5.1 and detailed in App. A.1. Implementation details for computing these metrics, along with any external libraries utilized, are provided to ensure transparency in our evaluation procedures. Extended experimental results, including standard deviations and additional visualizations, are included in App. B. Ablation studies and sensitivity analyses are presented to demonstrate the robustness of our methods. These additional results provide deeper insights into our findings and allow for a more thorough understanding of our approach.

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

## A    IMPLEMENTATION DETAILS

### A.1    METRICS

In this section, we describe the evaluation metrics used in our experiments in more detail. Following (Liu et al., 2024b), we use CLIP-IQAC, which calculates the CLIP score difference between "a good photo of [class]" and "a bad photo of [class]". For calculating IMS-VGGNet, we leverage the VGGNet in the DeepFace library for face recognition and face embedding extraction (Serengil & Ozpinar, 2021). For IMS-IP, we leverage *antelopev2* model from InsightFace library (Deng et al., 2020) following IP-adapter (Ye et al., 2023). We report the weighted average of them with a weighting factor on IMS-IP as 70% since we find it yields a more stable evaluation with IMS-VGG as 30%. We compute all the mean scores for all generated images and instances. For the instance $i$ and its $j$-th metric, its $k$-th observation value is defined as $m_{i,j,k}$. For the $j$-th metric, the mean value is obtained with $\sum_{i,k} m_{i,j,k}/(N_i N_k)$, where $N_i$ is the instance number for that particular dataset, and $N_k$ is the image generation number.

### A.2    HARDWARE AND TRAINING DETAILS

**Hardware Details.** All the experiments are conducted on an Ubuntu 20.04.6 LTS (focal) environment with 503GB RAM, 10 GPUs (NVIDIA® RTX® A5000 24GB), and 64 CPU cores (Intel® Xeon® Silver 4314 CPU @ 2.40GHz). Python 3.9.18 and Pytorch 1.13.1 are used for all the implementations. Please refer to the supplementary material for the code and environment setup.

**Training and Inference Settings.** The Stable Diffusion (SD) v2-1-base (Rombach et al., 2022) is used as the model backbone. For Dreambooth training, we conduct full fine-tuning, which includes both the text-encoder and U-Net model with a constant learning rate of $5 \times 10^{-7}$ and batch size of 2 for 1000 iterations in mixed-precision training mode. We use the 8-bit Adam optimizer with $\beta_1 = 0.9$ and $\beta_2 = 0.999$ under bfloat16-mixed precision and enable the xformers for memory-efficient training. For calculating prior loss, we use 200 images generated from Stable Diffusion v2-1-base with the class prompt "`a photo of a [class norn]`". The weight for prior loss is set to 1. For the evaluation phase, we set the inferring steps as 100 with prompts "a photo of sks person" and "a smiling photo of sks person" during inference to generate *16* images per prompt. For all the settings, the classifier-free guidance Ho & Salimans (2022) is turned on by default with a guidance scale of 7.5. For the implementation of baseline methods, please refer to App. D.

## B    MORE EXPERIMENTAL RESULTS

### B.1    QUALITY SCORE CURVE DURING TRAINING

We present the LIQE (Zhang et al., 2023a) quality score curve during fine-tuning under different settings, including clean training, vanilla training on perturbed data, training with CDL, and training with CodeSR+CDL in Figure 6. This curve illustrates the evolution of image quality throughout the training process. As evident from the figure, our proposed decoupled learning (CDL) approach significantly enhances the quality compared to the case with perturbations. Moreover, when we combine CDL with input purification (CodeSR + CDL), the model achieves quality performance comparable to clean-level training. These results further validate the effectiveness of our proposed method in defending against adversarial perturbations and maintaining high-quality outputs in PDMs.

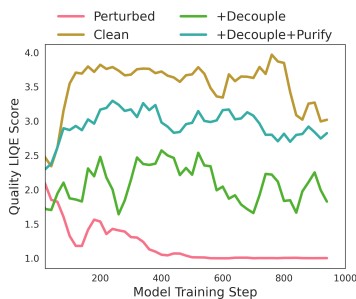

Figure 6: LIQE quality score of $\mathcal{V}*$.

### B.2    LEARNED CONCEPTS VISUALIZATION

To visually demonstrate our method's effectiveness, Fig. 7 compares the concept extraction results from trained models with vanilla training, CDL, and CodeSR+CDL. We extract three concepts from

the trained models, including the instance concept, instance+class concept, and decoupled noise concept. The third one aims to visualize the noise pattern from the perturbed data that we seek to decouple. From the figure, we can see that CDL helps the model learn the correct concept-image correlations while adding CodeSR, which further improves the generation quality. Interestingly, we find that the learned noise concept in CDL-based training matches the pattern of the one falsely linked by the personalized concept in vanilla training. We present more results supporting this in Fig. 9. This validates the effectiveness of our method in learning the correct concept-image correlations and decoupling the noise concept. Furthermore, from Fig. 7, we find that adding input purification (CodeSR) greatly boosts generation quality. Under the purification case, the contribution of CDL is more about decoupling the left-over background artifacts from the personalized concept.

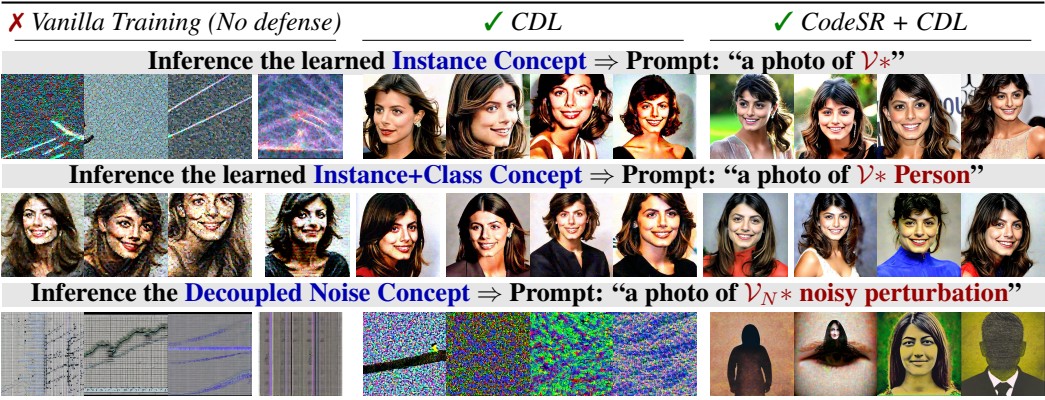

Figure 7: Concept extraction with three different prompts from the trained model with vanilla training, CDL, and CodeSR+CDL. Results show that CDL alone helps models learn the right correlations, and adding input purification further boosts the generation quality.

### B.3 CONTRASTIVE DECOUPLING LEARNING WITH DIFFERENT NOISE TOKENS.

To investigate the effect of using our CDL with different noise tokens, we additionally present results in Tab. 5. As we can see, setting the right noise tokens is crucial for the performance. Among the 15 noise tokens we tested, we found that *"t@j noisy pattern"* yielded the best overall performance under our setting. Future works can be conducted using automatic noise prompt searching. Another direction is to study visualization of the learned pattern for each noise prompt setting for a deeper understanding of the underlying concept learning process.

### B.4 MORE RESULTS ON PURIFICATION FAITHFULNESS

In addition to the LPIPS used in the paper, we provide purification results using other similarity metrics, including Structural Similarity (SSIM), Multi-Scale Structural Similarity (MS-SSIM), and Peak Signal-to-Noise Ratio (PSNR). The results are presented in Tab. 7, which demonstrate that our purification variants are consistently superior to previous state-of-the-art purification approaches.

### B.5 LIMITATIONS DISCUSSION

**Limitations.** While our proposed defense framework demonstrates significant improvements over existing methods in enhancing the robustness of PDMs, there are certain areas that could be further explored. Our experiments are primarily conducted on the facial dataset VGGFace2. Although we have preliminary purification results indicating the applicability of our approach to other domains like artwork images from WikiArt, we have not extensively tested our method across a wide variety of protection techniques. Future work could investigate the generalizability of our method to different types of images and subjects to further validate its effectiveness. Additionally, the integration of data purification and contrastive decoupling learning introduces some additional computational steps during the training process. This may slightly increase the training time compared to standard training procedures. However, we believe that this is a reasonable trade-off given the substantial benefits in

Table 5: Performance comparison of models trained with our Contrastive Decoupling Learning (CDL) using various noise tokens. Results are shown for seven evaluation metrics across different noise token choices. Higher scores indicate better performance. The experiment uses a single random instance from the VGGFace2 dataset, protected by MetaCloak. IMS and Q are our main metrics, while $\text{IMS}_{\text{VGG}}$, $\text{IMS}_{\text{IP}}$, LIQE, and CLIP-IQAC provide additional insights into model performance.

| Noise Tokens $\mathcal{V}_N *$ | IMS ↑ | Q ↑ | Avg. ↑ | $\text{IMS}_{\text{VGG}}$ ↑ | $\text{IMS}_{\text{IP}}$ ↑ | LIQE ↑ | CLIP-IQAC ↑ |
|---|---|---|---|---|---|---|---|
| *t@j noisy pattern* | -0.226 | 0.156 | -0.035 | -0.265 | -0.209 | 3.313 | 0.460 |
| *xjy image imperfection* | -0.331 | -0.130 | -0.230 | -0.511 | -0.253 | 2.740 | 0.324 |
| *xjy visual interference* | -0.539 | 0.014 | -0.263 | -0.513 | -0.550 | 3.028 | 0.130 |
| *xjy visual distortion* | -0.336 | -0.204 | -0.270 | -0.284 | -0.357 | 2.591 | 0.149 |
| *xjy image artifact* | -0.159 | -0.445 | -0.302 | -0.423 | -0.045 | 2.110 | 0.277 |
| *xjy digital glitch* | -0.294 | -0.378 | -0.336 | -0.448 | -0.227 | 2.243 | 0.328 |
| *UNKNOWN face degradation* | -0.197 | -0.476 | -0.337 | -0.296 | -0.155 | 2.047 | 0.520 |
| *xjy image disturbance* | -0.328 | -0.366 | -0.347 | -0.424 | -0.287 | 2.268 | 0.225 |
| *xjy image corruption* | -0.449 | -0.248 | -0.349 | -0.477 | -0.437 | 2.504 | 0.104 |
| *xjy image degradation* | -0.345 | -0.410 | -0.377 | -0.286 | -0.370 | 2.181 | 0.110 |
| *UNKNOWN noisy pattern* | -0.431 | -0.419 | -0.425 | -0.244 | -0.512 | 2.161 | -0.020 |
| *xjy visual anomaly* | -0.389 | -0.534 | -0.461 | -0.453 | -0.361 | 1.932 | 0.126 |
| *XX noisy artifact* | -0.324 | -0.626 | -0.475 | -0.143 | -0.401 | 1.749 | -0.096 |
| *xjy visual noise* | -0.578 | -0.475 | -0.526 | -0.288 | -0.702 | 2.050 | -0.060 |
| *bhi noisy perturbation* | -0.494 | -0.727 | -0.610 | -0.369 | -0.547 | 1.546 | -0.242 |

Table 6: The full results with standard deviations of different purification methods under different protective perturbations. The best performances are in **bold**, and second runners are shaded in gray. $*$ denotes improvement that passes the Wilcoxon signed-rank significance test with $p \leq 0.01$.

| Methods | FSMG | | ASPL | | EASPL | | MetaCloak | | AdvDM | | PhotoGuard | | Glaze | |
|---|---|---|---|---|---|---|---|---|---|---|---|---|---|---|
| | IMS ↑ | Q ↑ | IMS | Q | IMS | Q | IMS | Q | IMS | Q | IMS | Q | IMS | Q |
| Clean | -0.13 ± 0.04 | 0.15 ± 0.08 | -0.13 ± 0.04 | 0.15 ± 0.08 | -0.13 ± 0.04 | 0.15 ± 0.08 | -0.13 ± 0.04 | 0.15 ± 0.08 | -0.13 ± 0.04 | 0.15 ± 0.08 | -0.13 ± 0.04 | 0.15 ± 0.08 | -0.13 ± 0.04 | 0.15 ± 0.08 |
| Perturbed | -0.43 ± 0.54 | -0.54 ± 0.25 | -0.67 ± 0.46 | -0.52 ± 0.41 | -0.62 ± 0.46 | -0.50 ± 0.40 | -0.35 ± 0.58 | -0.53 ± 0.28 | -0.27 ± 0.54 | -0.36 ± 0.30 | -0.18 ± 0.54 | -0.24 ± 0.27 | -0.28 ± 0.59 | -0.28 ± 0.33 |
| Gaussian F. | -0.19 ± 0.57 | -0.55 ± 0.29 | -0.20 ± 0.56 | -0.57 ± 0.20 | -0.17 ± 0.56 | -0.58 ± 0.23 | -0.07 ± 0.54 | -0.63 ± 0.15 | -0.11 ± 0.54 | -0.57 ± 0.24 | -0.23 ± 0.56 | -0.53 ± 0.26 | -0.18 ± 0.53 | -0.54 ± 0.25 |
| JPEG | -0.15 ± 0.60 | -0.41 ± 0.33 | -0.21 ± 0.62 | -0.52 ± 0.25 | -0.27 ± 0.62 | -0.50 ± 0.24 | -0.34 ± 0.63 | -0.38 ± 0.41 | -0.15 ± 0.62 | -0.02 ± 0.39 | -0.13 ± 0.62 | 0.07 ± 0.36 | -0.19 ± 0.57 | -0.03 ± 0.48 |
| TVM | -0.12 ± 0.48 | -0.65 ± 0.21 | -0.16 ± 0.53 | -0.66 ± 0.23 | -0.10 ± 0.49 | -0.67 ± 0.20 | -0.11 ± 0.49 | -0.69 ± 0.12 | -0.12 ± 0.50 | -0.65 ± 0.22 | -0.15 ± 0.54 | -0.64 ± 0.20 | -0.11 ± 0.48 | -0.66 ± 0.20 |
| PixelDiffPure | -0.41 ± 0.45 | -0.57 ± 0.17 | -0.43 ± 0.47 | -0.54 ± 0.21 | -0.57 ± 0.49 | -0.61 ± 0.16 | -0.28 ± 0.44 | -0.58 ± 0.22 | -0.40 ± 0.51 | -0.55 ± 0.21 | -0.25 ± 0.51 | -0.55 ± 0.15 | -0.41 ± 0.44 | -0.59 ± 0.17 |
| L.DiffPure-∅ | -0.07 ± 0.50 | -0.47 ± 0.29 | -0.36 ± 0.49 | -0.59 ± 0.21 | -0.22 ± 0.58 | -0.49 ± 0.26 | -0.52 ± 0.46 | -0.43 ± 0.29 | -0.55 ± 0.45 | -0.24 ± 0.38 | -0.12 ± 0.48 | -0.40 ± 0.28 | -0.38 ± 0.45 | -0.42 ± 0.27 |
| L.DiffPure | -0.25 ± 0.48 | -0.45 ± 0.30 | -0.31 ± 0.51 | -0.61 ± 0.20 | -0.30 ± 0.54 | -0.46 ± 0.32 | -0.31 ± 0.46 | -0.51 ± 0.22 | -0.57 ± 0.43 | -0.30 ± 0.34 | -0.25 ± 0.48 | -0.47 ± 0.21 | -0.41 ± 0.46 | -0.47 ± 0.27 |
| DDSPure | -0.15 ± 0.61 | -0.34 ± 0.23 | -0.05 ± 0.59 | -0.38 ± 0.19 | -0.08 ± 0.54 | -0.39 ± 0.20 | -0.16 ± 0.59 | -0.49 ± 0.23 | -0.19 ± 0.59 | -0.43 ± 0.23 | -0.12 ± 0.59 | -0.37 ± 0.21 | -0.22 ± 0.55 | -0.41 ± 0.24 |
| GrIDPure | -0.10 ± 0.59 | -0.20 ± 0.23 | -0.21 ± 0.58 | -0.16 ± 0.23 | -0.13 ± 0.55 | -0.25 ± 0.25 | -0.23 ± 0.52 | -0.25 ± 0.25 | -0.09 ± 0.54 | -0.18 ± 0.26 | -0.03 ± 0.59 | -0.22 ± 0.26 | -0.24 ± 0.56 | -0.13 ± 0.28 |
| IMPRESS | -0.15 ± 0.58 | -0.53 ± 0.24 | -0.16 ± 0.60 | -0.49 ± 0.31 | -0.29 ± 0.60 | -0.64 ± 0.13 | -0.34 ± 0.58 | -0.29 ± 0.30 | -0.34 ± 0.56 | -0.34 ± 0.31 | -0.16 ± 0.58 | -0.21 ± 0.28 | -0.10 ± 0.59 | -0.43 ± 0.25 |
| Ours | **0.23*** ± 0.47 | **0.65*** ± 0.21 | **0.09** ± 0.48 | **0.62*** ± 0.15 | **0.09*** ± 0.49 | **0.63*** ± 0.19 | **0.38*** ± 0.38 | **0.58*** ± 0.27 | **0.29*** ± 0.44 | **0.67*** ± 0.20 | **0.24*** ± 0.49 | **0.63*** ± 0.19 | **0.31*** ± 0.43 | **0.66*** ± 0.25 |

Table 7: Purification faithfulness under various similarity metrics.

| Settings | LPIPS ↓ | SSIM ↑ | MS_SSIM ↑ | PSNR ↑ | Avg(IMS,Q) ↑ |
|---|---|---|---|---|---|
| IMPRESS | 0.451 | 0.761 | 0.903 | 49.294 | -0.63 |
| DDSPure | 0.384 | 0.805 | 0.873 | 46.948 | -0.65 |
| GrIDPure | 0.429 | 0.754 | 0.794 | 41.976 | -0.48 |
| L.DiffPure-∅ | 0.450 | 0.676 | 0.732 | 43.551 | -0.82 |
| Code ✓ + SR ✓ | **0.271** | 0.824 | 0.925 | 49.937 | **0.385** |
| Code ✓ + SR ✗ | 0.231 | **0.891** | **0.952** | **52.49** | 0.339 |
| Code ✗ + SR ✓ | 0.270 | 0.790 | 0.923 | 49.591 | 0.225 |

| Noisy Generation | LatentDiffPure-∅ | LatentDiffPure | DDSPure | GrIDPure | IMPRESS | CodeSR |
|---|---|---|---|---|---|---|

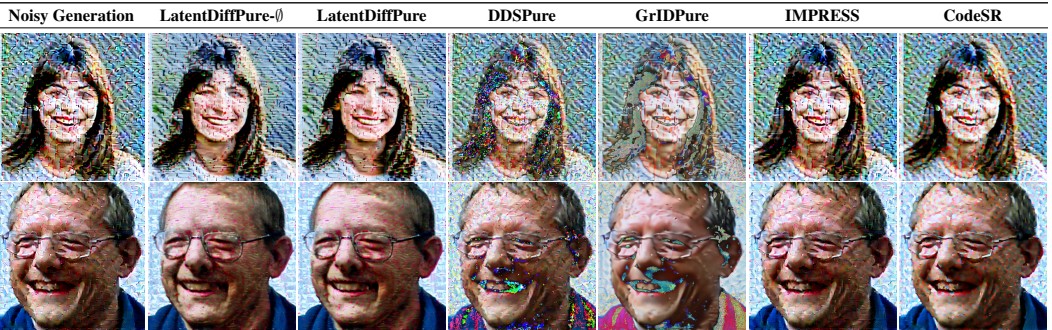

Figure 8: Visualization of post-hoc purification on noisy outputs of PDMs trained without input cleaning on protected images.

terms of robustness and generation quality that our method provides. While our framework demonstrates robustness against various adaptive perturbations, we acknowledge that more sophisticated protection techniques may emerge. For instance, our red-teaming setup currently focuses on noise-based protective perturbations, but object-embedded perturbations (Zhu et al., 2024) could potentially resist our noise-concept-based CDL prompt design. Additionally, to counter our purification pipeline, future protection techniques could explore more advanced ensemble methods (Chen et al., 2022).

**Discussion on Broader Impact.** Our work on red-teaming existing protective perturbations raises ethical considerations, particularly regarding privacy and intellectual property rights. While our methods could potentially compromise images protected by existing protective perturbations, we believe that the benefits of this research outweigh the potential risks. First, our research helps prevent a false sense of security by revealing limitations in existing protective measures. This transparency enables portrait owners and artists to make more informed decisions about protecting their content. Furthermore, the insights gained from our analysis can inform the development of next-generation protection techniques that are more resilient against sophisticated red-teaming, thereby strengthening privacy and copyright safeguards in the long term.

### B.6 PURIFICATION ON NOISY OUTPUTS

We additionally investigate whether post-hoc purification can effectively clean up the noisy outputs generated by PDMs trained without any defense. In the pixel domain, we observe that these generated images contain significant distortions manifesting as mosaic-like patterns and irregular fragmentation overlaid on the person's identity. As shown in Fig. 8, applying various state-of-the-art purification methods as denoisers fails to effectively remove these semantic distortions, indicating that once the model learns to generate distorted outputs, simple post-processing cannot restore clean image quality.

## C CAUSAL ANALYSIS OF LEARNING PERSONALIZED DIFFUSION MODELS ON PERTURBED DATA

### C.1 CONSTRUCTION OF THE CAUSAL GRAPH WHEN LEARNING PDMS ON PERTURBED DATA

To understand how protective perturbations lead to shortcut learning in PDMs, we construct a Structural Causal Model (SCM) that captures the learned causal relationships between the variables

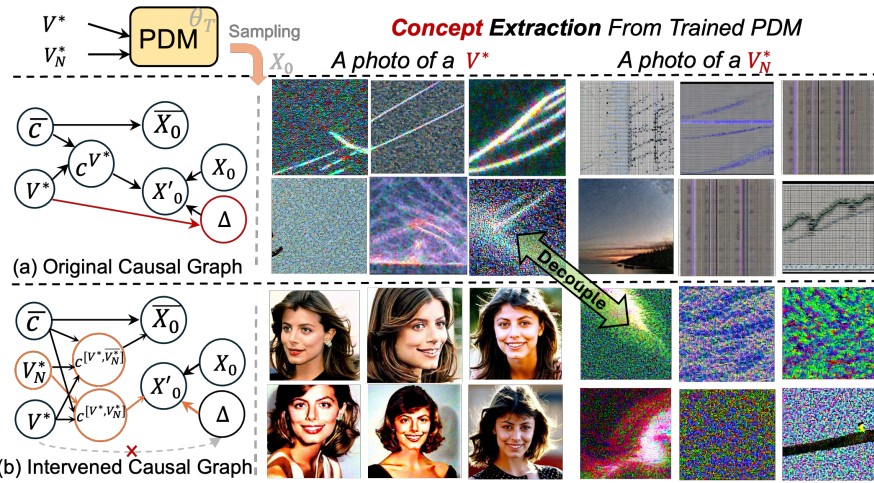

Figure 9: More visualization on the learned personalized and noise concepts from trained models with and without CDL. With concept extraction, we examine that using CDL successfully decouples the original noise pattern spuriously linked to personalized tokens $\mathcal{V}^*$ to the noise tokens $\mathcal{V}_N^*$.

involved in the fine-tuning process. The variables in our SCM are defined as follows: $X_0$ represents the original clean images representing the true concept; $\Delta$ denotes the protective perturbations added to the images; $X_0' = X_0 + \Delta$ are the perturbed images used for fine-tuning; $c$ represents class-specific textual prompts without the unique identifier (e.g., "a photo of a person"); $\mathcal{V}^*$ is the unique identifier token used in personalized prompts (e.g., "sks"); $c^{\mathcal{V}^*} = c \oplus \mathcal{V}^*$ denotes the personalized textual prompts combining $c$ and $\mathcal{V}^*$; $\theta_T$ represents the model parameters after being fine-tuned. The structural equations governing the relationships in our SCM are as follows: (1) Perturbed Images: $X_0' = X_0 + \Delta$, where $X_0'$ represents the perturbed images, $X_0$ the original clean images, and $\Delta$ the protective perturbations. (2) Model Fine-tuning: $\theta_T = f_\theta(\theta_0, X_0', c^{\mathcal{V}^*}, \bar{X}_0, \bar{c})$, where $\theta_T$ represents the fine-tuned model parameters, $\theta_0$ the initial model parameters, $c^{\mathcal{V}^*}$ the personalized text prompts, $\bar{X}_0$ and $\bar{c}$ the image and prompt of class-specific dataset to help model maintain class prior. For our case of fine-tuning on human portrait, the $\bar{X}_0$ is the person images from different identities, and $\bar{c}$ is set as "a photo of a person". After $\theta_T$ has been fine-tuned, it learns the latent causal relationship $\mathcal{V}^*$ $\rightarrow X_0'$ with conditioning mechanism through prompt-image association.

Based on these equations, we construct a causal graph shown in Fig. 2 (a) and Fig. 9 (a) following the conventions in causal inference. In the graph, we define each node to represent one of the elements for the learned causation: independent variables (i.e., text prompts, and unique identifier), dependent variables (i.e., perturbed identity images, general face images), or intermediate variables like prompt composited. We define each edge to represent the causal unidirectional dependency between the variables. For those prompt composition edges, the relationship is simply the concatenation operation in the textual space. For those prompt-image association edges, the relationship is defined as the causation learned by the model $\theta_T$. For the edges between $\Delta$ and $X_0'$, it is defined as the direct effect of the perturbations on the original clean images, $X_0' = X_0 + \Delta$. Similar to the confounder in causal inference, we can see from the graph that the perturbation $\Delta$ induces a shortcut connection from the unique identifier $\mathcal{V}^*$ to the noisy concept $\Delta$. Note that in the context of backdoor learning through causal inference, such confounder is termed trigger or backdoor variable (Zhang et al., 2023c; Liu et al., 2024a). Different from the backdoor scenario, our case of protective perturbation introduces confounding variables on the learning target side instead of on the input side in backdoor attacks.

## C.2 CONNECTION OF OUR DESIGN TO CAUSAL INTERVENTION

Our red-teaming strategies can be interpreted as interventions that modify the causal graph to weaken or eliminate the undesired shortcut connection between $\mathcal{V}^*$ and the noisy concept $\Delta$.

**Input Purification** aims to mitigate the effect of adversarial perturbations, but it's important to note that this process is not perfect. In the context of our causal model, we can represent this imperfect purification as:

$$X_0' = X_0 + \Delta \rightarrow X_0' = X_0 + \Delta_r, \tag{7}$$

where $\Delta_r$ represents the residual perturbations after purification, with $\|\Delta_r\| \ll \|\Delta\|$. This intervention partially weakens the path from $\Delta$ to $X_0'$, and consequently to $\theta_T$ and $X_0$. The distribution of the model outputs given this imperfect purification can be expressed as:

$$P(X_0 \mid \mathrm{do}(\Delta = \Delta_r), x_0, c^{\mathcal{V}^*}). \tag{8}$$

While input purification reduces the influence of adversarial perturbations on the fine-tuning process and subsequent image generation, it does not completely eliminate the shortcut learning problem. This limitation motivates the need for additional strategies to mitigate shortcut learning further.

**Contrastive Decoupling Learning (CDL)** intervenes on the potential shortcut $\mathcal{V}^* \rightarrow \Delta$ by introducing a noise identifier $\mathcal{V}_N^*$. By augmenting the instance prompts to include a noise identifier (e.g., "a photo of $\mathcal{V}^*$ with $\mathcal{V}_N^*$ noisy pattern") and augmenting the class prompts to exclude it (e.g., "a photo of a person without $\mathcal{V}_N^*$ noisy pattern"), CDL encourages the model to disentangle the learning of the personalized concept from the noise patterns. Specifically, inherently, the model learns two clearer associations, $\mathcal{V}_N^* \rightarrow \Delta$, and $\mathcal{V}^* \rightarrow X_0$. By defining the variable that represents "without $\mathcal{V}_N^*$" as $\bar{\mathcal{V}}_N^*$, we can further compose a intervened causal graph as shown in Fig. 2 (b) and Fig. 9 (b). We use orange color to highlight the main intervened node and edges in the graph. From the results in Fig. 9, we see that the decoupling process enables the model to learn two concepts separately, including the personalized concept and the noisy pattern. Furthermore, during the sampling stage, we apply classifier-free guidance (CFG) to further improve the quality of the generated images. It modifies the generation process by incorporating negative prompts during inference, thereby adjusting the output equation to $g'(\theta_T, c^{[\mathcal{V}^*, \bar{\mathcal{V}}_N^*]}, c_{\mathrm{neg}})$, where $g'$ is the modified generation function and $c_{\mathrm{neg}}$ are negative prompts (e.g., "noisy, abstract, pattern, low quality"). We guide the model to generate images that don't contain any noisy pattern associated with $\mathcal{V}_N^*$ in the prompt input. This step acts as an intervention on the generation mechanism, reducing the influence of any residual associations between $\Delta$ and the outputs. Although it is more of a practical adjustment than a formal causal intervention, it helps steer the model toward generating high-quality images that reflect the clean personalized concept.

In summary, by combining these strategies, we provide a comprehensive approach to mitigate shortcut learning in PDMs. Input purification directly removes the influence of adversarial perturbations, and our CDL further reduces potential left-over spurious associations during training. In the final sampling phase, we use CFG to guide the model generation process by discouraging undesired artifacts and encouraging the generation of high-quality images that reflect the personalized concept.

## D   IMPLEMENTATION OF BASELINES

### D.1   PURIFICATION METHODS

We implement two classes of purification approaches; the first ones are model-free and operate with certain image processing algorithms, such as Gaussian Filtering, total variation minimization (TVM), and JPEG compression. Despite the simplicity, researchers found that these approaches can achieve non-trivial defense performance against adversarial attacks (Liang et al., 2023), availability attacks (Liu et al., 2024b; Van Le et al., 2023), and more general data poisoning attacks (Huang et al., 2020). Another line of approach is based on powerful diffusion probabilistic models, which have a strong ability to model real-world data distribution and also show potential in being leveraged for zero-shot purifiers Shi et al. (2024); Zhao et al. (2024a); Carlini et al. (2022); Cao et al. (2024). We include a wide range of SoTA diffusion-based purification approaches that are designed for the protective perturbation specifically, including GrIDPure (Zheng et al., 2023), IMPRESS (Cao et al., 2024), or those are proposed for more general adversarial perturbation (Nie et al., 2022; Carlini et al., 2022), including DiffPure (Nie et al., 2022) (with pixel-space diffusion models or latent-space diffusion models), DDS-based purification (DDSPure) (Carlini et al., 2022; Hu et al., a).

**1. Gaussian Filtering.** Gaussian Filtering is a well-known image-processing technique used to reduce image noise and detail by applying a Gaussian kernel. The high-frequent part in adversarial

perturbation can be smoothed after filtering. The kernel size is set as 5 following (Van Le et al., 2023).

**2. Total Variation Minimization (TVM) (Wang et al., 2020)** The main idea of TVM is to conduct image reconstruction based on the observation that the benign images should have low total variation. We implemented the TVM defense in the following steps: we first resized the instance image to $64^2$ pixels, applied a random dropout mask with a 2% pixel dropout rate, and solved a TVM optimization problem. The optimization aims to reconstruct the image by minimizing the difference between the original and reconstructed images while enforcing smoothness through the total variation term: $\min_Z ||(1-X) \odot (Z-x)||_2 + \lambda_{TV} TV_2(Z)$. After optimization, the reconstructed image is reshaped back to $64^2$ and then upsampled to $512^2$ through two SR steps with a middle resizing process.

**3. JPEG Compression.** It involves transforming an image into a format that uses less storage space and reduces the image file's size. We set the JPEG quality to 75 following (Liu et al., 2024b).

**4. DiffPure (Nie et al., 2022).** Diffusion Purification (DiffPure) first diffuses the adversarial example with a small amount of noise given a pre-defined timestep $t$ following a forward diffusion process, where the adversarial noise is smoothed and then recovers the clean image through the reverse generative process. Depending on the type of diffusion model used, this simple yet effective approach can be adapted into two versions: PDM-based DiffPure and LDM-based DiffPure. In our implementation, we term the PDM-based DiffPure as *PixelDiffPure* for short and leverage `256x256_diffusion_uncond` pre-trained on ImageNet released in the `guided-diffusion` following common practice. For the LDM-based DiffPure, we term it as LatentDiffPure since the diffusion process is conducted in latent space and leverage Stable Diffusion v1-4 (Rombach et al., 2022) for its superior performance. Since the SD model has the ability to input additional text prompts during the purification process, we investigate two variants with and without the usage of purified text prompting. For LatentDiffPure-$\emptyset$, we set the text to null, while for LatentDiffPure, we set it as "`a photo of [class_name], high quality, highres`".

**5. DDSPure (Carlini et al., 2022).** Similar to DiffPure (Nie et al., 2022), the main idea behind Diffusion Denoised Smoothing (DDS) is to find an optimal timestamp that can maximally remove the adversarial perturbation via the SDEdit process (Meng et al., 2021). Given smoothing noise level $\delta$, the optimal timestamp $t^*$ is computed via, $t^* = \frac{1-\bar{\alpha}_t}{\bar{\alpha}_t} = \sigma^2$. Following common practice, we leverage the pretrained diffusion model on ImageNet released in the `guided-diffusion`. Specifically, the `256x256_diffusion_uncond` is used as a denoiser. To resolve the size mismatch, we resize the images to fit the model input and resize the image size back after purification. And we clip $t^*$ when it falls outside the sampling step range of $[0, 1000]$.

**4. GrIDPure (Zheng et al., 2023).** GrIDPure notices that for purification in defending protective perturbation, conducting iterative DiffPure with small steps can outperform one-shot DiffPure with larger steps. Furthermore, it suppresses the generative nature during diffusion purification by additionally splitting the image into multiple small grids that are separately processed with a final merging process. This allows the model to focus more on purifying those perturbed textures and curves in the image without mistakenly affecting the overall structure, thus preserving the faithfulness of purification.

---

**Algorithm 2** GrIDPure

---

**Input:** Input image $x_0$, number of iterations $N$, time-stamp $t$, grid size $g$, stride $s$, merging weight $\gamma$
**Output:** Purified image $x_N$
1: **for** $i = 0$ to $N-1$ **do**
2:     Split $x_i$ into grids of size $g \times g$ with stride $s$
3:     **for** each grid $x_{i,j}$ **do**
4:         Apply DiffPure with time-stamp $t$ to obtain $\tilde{x}_{i,j}$
5:     **end for**
6:     Merge all $\tilde{x}_{i,j}$ to obtain $\tilde{x}_i$, averaging pixel values in overlapping regions
7:     $x_{i+1} = (1-\gamma) \cdot \tilde{x}_i + \gamma \cdot x_i$
8: **end for**
9: **return** $x_N$

---

Given an input image size of $512 \times 512$, we implement the GrIDPure algorithm as follows with the hyper-parameter recommended in the original paper. We first obtain multiple grids using a sliding window approach. The window size is $256 \times 256$, and the stride is $128$. For each $256 \times 256$ grid, we apply DiffPure with a time-stamp of $t = 10$. After all the grids are denoised, they are merged back into a single image. In the overlapping regions, the pixel values are averaged. Given $\gamma$ as $0.1$, the purified image is then obtained via a moving average with the original image,

$$\boldsymbol{x}_{i+1} = (1 - \gamma) \cdot \tilde{\boldsymbol{x}}_i + \gamma \cdot \boldsymbol{x}_i. \tag{9}$$

These steps constitute one iteration, and the algorithm is repeated for a total of $10$ iterations. We implement the GrIDPure algorithm following their official implementation.

**6. IMPRESS (Cao et al., 2024)** The key idea of IMPRESS is to conduct purification that ensures *latent consistency with visual similarity constraints*: (1) the purified image should be visually similar to the perturbed image, and (2) the purified image should be consistent upon an LDM-based reconstruction. To quantify the similarity condition, IMPRESS uses the LPIPS metric (Zhang et al., 2018), which measures the human-perceived image distortion between the purified image $\mathbf{x}_{\text{pur}}$ and the perturbed image $\mathbf{x}_{\text{ptb}}$. The loss is defined as $\max(\text{LPIPS}(\mathbf{x}_{\text{pur}}, \mathbf{x}_{\text{ptb}}) - \Delta_L, 0)$, where $\Delta_L$ is the perceptual perturbation budget. For the consistency condition, IMPRESS simplifies the loss by removing the diffusion process and defines it as $||\mathbf{x}_{\text{pur}} - \mathcal{D}(\mathcal{E}(\mathbf{x}_{\text{pur}}))||_2^2$, where $\mathcal{E}$ and $\mathcal{D}$ are the image encoder and decoder in the LDM, respectively. The final optimization problem combines the two losses:

$$\min_{\mathbf{x}_{\text{pur}}} ||\mathbf{x}_{\text{pur}} - \mathcal{D}(\mathcal{E}(\mathbf{x}_{\text{pur}}))||_2^2 + \alpha \cdot \max(\text{LPIPS}(\mathbf{x}_{\text{pur}}, \mathbf{x}_{\text{ptb}}) - \Delta_L, 0), \tag{10}$$

where $\alpha$ is a hyperparameter to balance the two losses, which is set as $0.1$. The optimization is solved with PGD (Madry et al., 2018) with Adam optimizer with lr of $0.001$, and the total iteration is set as $3000$.

### D.2 PROTECTIVE PERTURBATION METHODS

We test a wide range of protective perturbation approaches, including those that craft noise against fixed LDMs by exploiting the out-of-distribution adversarial vulnerability of DNNs (Liang et al., 2023; Liang & Wu, 2023; Xue et al., 2023; Salman et al., 2023; Shan et al., 2023), and those that jointly and alternatively learn the noise generator and perturbation (Van Le et al., 2023; Liu et al., 2024b; Xu et al., 2024), which show better protection capacity for the LDM fine-tuning settings (Kumari et al., 2023; Ruiz et al., 2023).

**Fully-trained Surrogate Model Guidance (FSMG).** Following (Shan et al., 2020; Yeh et al., 2020), FSMG employs a surrogate DreamBooth model with original parameters $\theta_{clean}$ fully finetuned on a small subset of clean samples $\mathcal{X}_A \subset \mathcal{X}$. We implement the subset with the same identity to maximize the protection capability. Using $\theta_{clean}$ as guidance, we find the optimal noise for each target image: $\delta^{*(i)} = \arg\max_{\delta^{(i)}} \mathcal{L}_{cond}(\theta_{\text{clean}}, x^{(i)} + \delta^{(i)})$, where $\mathcal{L}_{cond}$ is the conditional denoising loss. This encourages any DreamBooth model finetuned on the perturbed samples to deviate from $\theta_{clean}$ and generate low-quality images.

**Alternating Surrogate and Perturbation Learning (ASPL).** Since FSMG fails to effectively solve the underlying bi-level optimization, inspired by Huang et al. (2021), ASPL further alternates the training of the surrogate DreamBooth model with perturbation learning. The surrogate model $\epsilon_\theta$ is initialized with pre-trained weights. In each iteration, a clone $\epsilon'_{\theta'}$ is finetuned on clean reference data to simulate the learning trajectory on potential leaked clean data. This model is then used to expedite learning adversarial noises $\delta^{(i)}$ with denoising-error-maximization in the current loop. Finally, ASPL updates the actual surrogate model $\epsilon_\theta$ on the updated adversarial samples with gradient descent and proceeds to the next iteration. This procedure allows the surrogate model to mimic better the models trained by malicious DreamBooth users, as it is only trained on perturbed data.

**Ensemble-based ASPL (EASPL).** Since the model trainer's pre-trained text-to-image generator is often unknown, an improved approach is to use an ensemble (Cherepanova et al., 2021; Yang et al., 2021) of surrogate models finetuned from different pre-trained generators, which can lead to better transferability. We implement this approach with three surrogates. Besides, we follow the practice of a single model at a time in an interleaving manner to produce optimal perturbed data due to GPU memory constraints.

**MetaCloak.** Despite the effectiveness of perturbation crafted from noise-surrogate joint learning, studies find that these approaches lack robustness against simple data transformations such as minor Gaussian filtering. To address this issue, MetaCloak (Liu et al., 2024b) solves the underlying bi-level poisoning problem using a meta-learning framework with an additional transformation sampling process to craft transferable and robust perturbations. Incorporating an additional transformation process and a denoising-error maximization loss brings severe performance degradation in a generation.

**PhotoGuard.** PhotoGuard (Salman et al., 2023) mainly focuses on the setting of malicious editing where the diffusion models are fixed. It introduces two target-adversarial-perturbation-based (TAP-based) approaches: encoder attack and diffusion attack. The encoder attack adds a perturbation $\delta_{\text{enc}}$ to an image $\mathbf{x}$ such that the image encoder $\mathcal{E}$ produces a closer latent representation for $\mathbf{x} + \delta_{\text{enc}}$ and a target image $\mathbf{x}_{\text{target}}$. The diffusion attack crafts a perturbation $\delta_{\text{diff}}$ such that the LDM-reconstructed images based on the input are closer to some $\mathbf{x}_{\text{target}}$. The diffusion attack considers the whole LDM model with prompts, achieving better empirical performance but being less efficient compared to the encoder attack.

**GLAZE.** GLAZE (Shan et al., 2023) mainly focuses on artwork protection and aims to add perturbations to an artist's artworks such that LDMs cannot learn the correct style from the perturbed artworks. Similar to the TAP-based encoder attack in PhotoGuard, it first chooses a target style $T$ sufficiently different from the style of the original image $\mathbf{x}$. Then, it transfers $\mathbf{x}$ to the target style using a pre-trained style transfer model $\Omega$. Given the style-transferred image $\Omega(\mathbf{x}, T)$, GLAZE crafts the perturbation $\delta_{\text{GLAZE}}$ by minimizing the distance between the encodings of $\Omega(\mathbf{x}, T)$ and $\mathbf{x} + \delta$ while regularizing the perceptual distortion using LPIPS. This encourages LDMs to generate samples with the target style instead of the original style when learning from the perturbed images.

**AdvDM**. Different from the above targeting attack, AdvDM (Liang et al., 2023) is proposed to optimize the adversarial perturbation in an untargeted and denoising-error-maximizing way. In detail, instead of learning a perturbation over one single reserve process, AdvDM learns the Monte-Carlo estimation of adversarial perturbation by sampling across all $t$ to maximize the denoising loss.

# E    MORE RELATED WORK AND DISCUSSIONS

## E.1    BACKDOOR ATTACKS AND DEFENSES.

**Backdoor Attacks and Defenses in Diffusion Models.** Backdoor attacks have emerged as a critical security threat to deep learning models, where malicious actors inject hidden functionalities during training that can be triggered during inference to manipulate model outputs. In the context of diffusion models, recent works have demonstrated their vulnerability to backdoor attacks through various approaches. Chou et al. (2022) first showed how to engineer compromised diffusion processes during training for backdoor implantation. Following studies explored more sophisticated attack approaches: Chou et al. (2023) presented a unified framework for attacking both conditional and unconditional diffusion models, while Li et al. (2024b) developed invisible triggers to enhance attack stealthiness. For text-to-image models specifically, Zhai et al. (2023) demonstrated backdoor attacks through multimodal data poisoning, and Huang et al. (2023) exploited model personalization as an attack approach. Several defense mechanisms have been proposed, including textual perturbations (Chew et al., 2024), distribution shift-based detection (An et al., 2023), and unified defense frameworks like T2IShield (Wang et al., 2024b) and TERD (Mo et al., 2024). The rapid development of both attacks and defenses highlights the ongoing arms race in securing diffusion models against backdoor threats.

**Difference between Protective Perturbations and Backdoor Attacks.** While protective perturbations in our problem share many similarities with backdoor attacks, they are fundamentally different. First, while they are both targeting implant some hidden and spurious correlation during the model learning process, particle backdoor attacks in diffusion models mainly focus on injecting backdoor triggers into the textual prompt part, while protective perturbations only alter the target image side without any explicit textual trigger added. Second, the backdoor attacks usually seek to maintain the model performance on normal queries, while the goal of protective perturbations is to degrade the model's overall performance in generating the target identity. Thirdly, backdoor attacks in diffusion models usually focus on the optimization of the model itself $\theta$ instead of crafting the perturbation $\delta$ in the input side as protective perturbations do. The backdoored model is learned to balance the maintenance of utility on normal queries and attack successful rate on the trigger queries, while the

protective perturbation usually operates on the input side, seeking to find transferable and robust perturbation that can fool a wide range of surrogate models.

**Causality-based Backdoor Defense and Detection.** Recent works have started exploring causality-based approaches to defend against and detect backdoor attacks. From a causal perspective, backdoor attacks can be viewed as confounders that introduce spurious correlations between input features and model predictions. Early works focused on using causal inference to analyze the robustness and effectiveness of existing backdoor defenses (Qiu et al., 2022). More recent approaches leverage causal reasoning to develop new defense mechanisms. For example, Min et al. (2024) reveals that current safety purification methods are vulnerable to rapid re-learning of backdoor behavior and proposes Path-Aware Minimization to improve post-purification robustness. Khaddaj et al. (2023) shows that without structural information about training data distribution, backdoor attacks are indistinguishable from naturally occurring features and develop a new detection primitive based on the assumption that these attacks correspond to the strongest feature in training data. A recent black-box detection approach termed Causality-based Black-Box Backdoor Detection (CaBBD) (Hu et al., b) models backdoor attacks as confounders and uses counterfactual samples as interventions to distinguish backdoor samples from clean ones. By progressively adding noise to generate these counterfactuals, the method achieves strong detection performance while maintaining inference efficiency.

A notable recent work in this direction is the Causality-inspired Backdoor Defense (CBD) (Zhang et al., 2023c). CBD approaches the problem by modeling the backdoor attack as a confounder in a causal graph, where the attack creates spurious paths between input images and predicted labels. The key insight is that while humans can distinguish causal relations from statistical associations, deep learning models tend to learn both without discrimination. CBD proposes a novel defense framework that learns de-confounded representations through (1) intentionally training a model to capture backdoor correlations, (2) training a clean model that minimizes mutual information with the backdoored model's representations, and (3) employing information bottleneck and sample re-weighting strategies to help the clean model focus on causal effects. Another significant advancement in causality-based defense is the Front-door Adjustment for Backdoor Elimination (FABE) (Liu et al., 2024a). Unlike CBD which focuses on backdoor confounders, FABE introduces a novel front-door adjustment approach specifically designed for language models. The key innovation is using a defense language model to generate semantically equivalent texts that serve as front-door variables, effectively breaking the spurious correlations introduced by backdoor attacks. FABE operates without requiring knowledge of trigger types by leveraging three key components: (1) a module for sampling front-door variables through instruction-tuned language models, (2) a causal effect estimation module for front-door adjustment formula, and (3) a gradient-based optimization for the front-door variables.

**Comparison with Zhang et al. (2023c) and Liu et al. (2024a).** Our work and these works both leverage causality-based perspectives to defend or red-teaming the perturbation. However, the problem and techniques in our work are fundamentally different from these two works. First, in terms of the problem, CBD and FABE both focus on the classification task, either image classification or text classification, where the backdoor spurious path is established between the model input $X$ and class label prediction $Y$. For our task, we are tackling the personalized generation task, where the LDMs are fine-tuned to link a unique identifier $\mathcal{V}^*$ to a new subject concept $X_0$. In the backdoor attack case, the attacker aims to introduce a confounder $A$ variable at the input side to trigger certain label prediction $Y'$, while in our case, the image protector only modifies the learning target $X_0' = X_0 + \Delta$ but do not explicitly add any trigger at the input side, which serves as the confounder in backdoor attack case. Thus, considering the difference in the threat model, the defense techniques in backdoor case, such as CBD and FABE, focus more on removing the confounder in the input side, while the defense in our case focuses on the prediction side, by reinforcing the causal path between the unique identifier $\mathcal{V}^*$ and the clean target concept $X_0$.

Second, in terms of techniques, both CBD and FABE only focus on one perspective on causal intervention, while our work proposes a unified framework that conducts both do-calculus (i.e., removing the injected variable or purification) and decoupling learning. Specifically, CBD assumes that the correlations $A \rightarrow Y$ can be well captured by an early-stop model $f_B$, and CBD learns the clean model $f_C : X \rightarrow Y$ by minimizing the mutual information between the embedding from $f_B$ and $f_C$. Compared to this feature space decoupling learning, our work operates the prompt augmentation side, which can be more efficient and end-to-end. Specifically, we observe the fact that the class-specific image doesn't contain any perturbation, while the instance image might contain the perturbation. Thus, we introduce a new noise identifier $\mathcal{V}_N^*$ and append it to two different datasets

with different prefixes "with" and "without" to achieve contrastive decoupling learning without any need to access the model weights and tuning any early stopping hyper-parameters as in CBD.

Similar to the purification part in our work, FABE mainly focuses on conducting semantic denoising on the original textual input to approximately achieve the do-calculus from the causal intervention perspective. Specifically, FABE denoise the $X$ to semantically equivalent text $Z$, with a fine-tuned language model. The fine-tuned language model learns to rank that effective $Z$ that removes confounder $A$, i.e., the backdoor trigger. Then, the prediction is conducted via voting over a pool of sampled $Z$ to achieve a clean prediction of $Y$. Compared to FABE, our purification pipeline for protective perturbation is more direct and flexible, without the need to fine-tune an additional model. Meanwhile, FABE requires unrolling $B$ semantic candidates using beam search, which can be computationally expensive especially when context length $L$ is large. In contrast, we leverage off-the-shelf image restoration and super-resolution models to conduct one-shot efficient purification.

### E.2    OTHER DIRECTIONS TOWARD DATA COPYRIGHT PROTECTION

**Digital Watermarking.** Digital watermarking is one of the most widely adopted approaches for protecting the intellectual property rights of digital content. It involves embedding identifying information (watermarks) into the target data in a way that is difficult to remove while maintaining the utility of the data (Saini & Shrivastava, 2014). Recent advances in deep learning have led to more sophisticated watermarking techniques. For instance, Zhang et al. (2023b) proposed EditGuard, a versatile framework that enables both tamper localization and copyright protection through spatial watermark embedding. Saberi et al. (2024) introduced DREW, which leverages error-controlled watermarking for robust data provenance. The key challenges in watermarking include achieving robustness against various attacks while maintaining imperceptibility and data utility.

**Source Attribution.** Data attribution enables the data owners to trace and verify the influence of their data on model outputs, providing a crucial mechanism for intellectual property protection in the era of generative AI. Traditional approaches often relied on watermarking techniques (Cui et al., 2023; Peng et al., 2023), which can be fragile against model modifications. More recent work has focused on developing robust attribution methods that can withstand various transformations while maintaining high detection accuracy. For instance, Singla et al. (2023) proposed an efficient baseline using self-supervised learning features that achieves strong attribution performance with significantly reduced computational overhead compared to previous ensemble-based methods. Wang et al. (2023b) introduced a comprehensive framework for evaluating attribution methods in text-to-image models, considering the inherent uncertainty in the attribution process. To address the challenges of scalability and personalization, Li et al. (2024a) developed an integrated approach combining proactive watermarking with passive detection for tracing generated content back to its source. These advances in attribution technology not only protect intellectual property rights but also create incentives for content owners to share their data (Ren et al., 2024). The emergence of libraries like `dattri` (Deng et al., 2024b) has further standardized and simplified the implementation of attribution methods, making them more accessible to practitioners.

**Model Unlearning.** With increasing privacy concerns and regulations like GDPR's "right to be forgotten", model unlearning has emerged as a crucial technique for removing specific data points' influence from trained models (Liu et al., 2024c). Unlike traditional retraining approaches, efficient unlearning methods aim to selectively eliminate the impact of certain training samples while preserving model performance on remaining data. Recent works like Panda et al. (2024) proposed partially blinded unlearning from a Bayesian perspective, while Zhao et al. (2024b) developed pseudo-probability unlearning for efficient and privacy-preserving removal of training data influence.

**Membership Inference Attacks.** Membership inference attacks attempt to determine whether specific data points were used in training a model, posing privacy risks to training data (Shokri et al., 2016). To protect against such attacks, various defense mechanisms have been proposed. Bernau et al. (2019) investigated the effectiveness of differential privacy in preventing membership inference, while Shateri et al. (2023) focused on preserving privacy in GANs against these attacks. Recent work by Laszkiewicz et al. (2023) explored the connection between data watermarking and set-membership inference, highlighting the interplay between different protection mechanisms.

