# OpenReview forum: "Rethinking and Defending Protective Perturbation in Personalized Diffusion Models"
_ICLR.cc/2025/Conference — Submitted to ICLR 2025_

### Official Review · Reviewer_uRZe · 2024-10-28

**Soundness:** 2
**Presentation:** 1
**Contribution:** 2
**Rating:** 3
**Confidence:** 4

**Summary:**

This paper proposes viewing the fine-tuning process of Personalized Diffusion Models (PDMs) through the lens of shortcut learning, using causal analysis as motivation. The authors then introduce a defense framework designed to enable the model to correctly associate images with their original semantic meanings.

**Strengths:**

The paper provides preliminary experiments on CLIP, which help demonstrate the authors' ideas.
Personalized diffusion models present an interesting area for further exploration.

**Weaknesses:**

1. The paper lacks overall coherence, with some sections difficult to follow and, in some cases, contradictory. Additionally, several terms and graphs are missing clear definitions and explanations.

    1. Are "adversarial perturbations" and "protective perturbations" intended to be the same concept? The author seems to use these terms interchangeably; if they differ, please clarify each term carefully.
    2. In the introduction, the author presents multiple related works. It may be helpful to focus on those most relevant to the paper’s main motivation. Additionally, certain terms, such as "purification studies," would benefit from brief explanations—similar to the way "image purifications" is introduced on line 142.
    3. Several equations need further explanation, such as those on lines 178-179, regarding the function of an instance dataset and a class dataset. Additionally, the meaning of "r" on line 208 is unclear.

2.After reading the entire paper, I found it challenging to identify the specific question the author aims to address and the associated motivations. While the introduction attempts to outline these points, it is difficult to discern the relationship between the motivation and the problem being addressed. Additionally, there appears to be a disconnect between the problem definition in the introduction and the methods presented. Here are some specific suggestions for clarification:

    1. The introduction states, “The model trained on perturbed data will generate images that are poor in quality, and thus, unauthorized fine-tuning fails.” Does this imply that generating low-quality images of private content protects copyright and privacy? If so, why does the proposed method focus on enhancing image clarity for private content while defining it as a defense?
    2.The author mentions that shortcuts are key to avoiding the generation of private personal images. Given this, why does the method seem to eliminate these shortcuts?
    3.On line 46, adversarial perturbations are suggested as a means to protect users’ images from unauthorized personalized synthesis. However, line 100 describes an intention to "defend against" this. Could you clarify?
    4.Additionally, the highlighted question in the introduction, “How to design an effective, efficient, and faithful purification approach is still an open question,” lacks context. Although there is a mention of “Moreover, purification studies are also purposed to further break those protections” in the following sentence, there are no subsequent explanations, particularly concerning how this question connects with the paragraph's earlier discussion.
    5. In the end of introduction, it seems that the authors propose a new purify methods, "Our approach conducts comprehensive purification from three perspectives, including input image purification, contrastive decoupling learning with the negative token, and quality-enhanced sampling....". However, in the methods, the author says they propose a method to address the short cut learning...., which is a little bit confusing.

3. Minor: Although viewing fine-tuning from a causal effect and shortcut learning perspective is novel, it shares similarities with backdoor attacks. In the backdoor attack literature, several papers have employed causal graphs to analyze shortcut mechanisms.[1-3]

4. The causal graph is underexplained and possibly contains ambiguities. For example, the definitions of $\bar{C}$ and $\bar{x_o}$ are missing. While a brief introduction to the construction of the graph is provided, explanations of each node’s meaning and the meaning of the arrows are absent. Given that the causal graph is a key contribution, adding a paragraph to introduce and explain it in detail would be beneficial. The term "spurious path" may also be misapplied; in causal inference, this usually refers to a backdoor path between treatment and outcome. Since this doesn’t apply here, either avoid the term or define it within the paper’s context.

5. The causal graph may need structural revision. In causal inference, an arrow between A and B signifies that A causes B. However, in this graph, it seems that an arrow signifies containment rather than causation. I would suggest adhering closely to causal inference conventions and adjusting the graph accordingly.

[1]Zhang Z, Liu Q, Wang Z, et al. Backdoor defense via deconfounded representation learning[C]//Proceedings of the IEEE/CVF Conference on Computer Vision and Pattern Recognition. 2023: 12228-12238.

[2]Liu Y, Xu X, Hou Z, et al. Causality Based Front-door Defense Against Backdoor Attack on Language Models[C]//Forty-first International Conference on Machine Learning.

[3]Hu M, Guan Z, Zhou Z, et al. Causality-Based Black-Box Backdoor Detection[J].

**Questions:**

See weakness

---

> ### Author Response · Authors · 2024-11-20
> **Part 1**
>
> We thank the reviewer for their valuable and detailed feedback. Based on the reviewer's suggestions, we have revised our manuscript accordingly. We acknowledge that some aspects of our motivation and setup may not have been clearly communicated, and we would like to clarify and address each of the concerns raised, as outlined below.
>
> > Q1: The paper lacks overall coherence, with some sections difficult to follow and, in some cases, contradictory. Additionally, several terms and graphs are missing clear definitions and explanations.
>
> **Response:** We have made significant efforts to improve the presentation in the revised manuscript, ensuring that our main motivations and contributions are clearly articulated. Below, we address each of the specific points raised.
>
> > Q1.1. Are "adversarial perturbations" and "protective perturbations" intended to be the same concept? The author seems to use these terms interchangeably; if they differ, please clarify each term carefully.
>
> **Response:** We apologize for the confusion caused by the interchangeable use of “adversarial perturbations” and “protective perturbations.” As per GrIDPure (Zhao et al., 2024) [1], these terms refer to the same concept but from different perspectives. “Adversarial perturbations” emphasize the perturbations’ disruptive effect on the model fine-tuning process from the model trainer’s viewpoint, while “protective perturbations” highlight their role in safeguarding portrait owners’ images from unauthorized synthesis. **To avoid confusion, we have standardized the terminology in the revised manuscript to consistently use “protective perturbations.”**
>
> > Q1.2. In the introduction, the author presents multiple related works. It may be helpful to focus on those most relevant to the paper’s main motivation. Additionally, certain terms, such as "purification studies," would benefit from brief explanations—similar to the way "image purifications" is introduced on line 142.
>
> **Response:** In the revised manuscript, we have refined the introduction to focus on the most relevant related works that align with our main motivation, particularly discussing the limitations of IMPRESS and GrIDPure. We have also provided brief explanations and appropriate citations for terms like “purification studies” to enhance clarity.
>
> > Q1.3. Several equations need further explanation, such as those on lines 178-179, regarding the function of an instance dataset and a class dataset. Additionally, the meaning of "r" on line 208 is unclear.
>
> **Response:** **We have added detailed explanations for these terms in the revised manuscript.** Specifically, as outlined in our preliminary section, in DreamBooth, the **instance dataset** contains images of the specific subject (e.g., portraits of a particular person) that the model is intended to learn. To prevent “language drift,” where the model might incorrectly associate the class name (e.g., “person”) exclusively with the subject instance, DreamBooth also employs a **class dataset** comprising images of the same class but with different identities. This helps the model retain general class-specific knowledge during fine-tuning. The weighted denoising losses for these datasets are presented in Equations 1 and 2 in the paper. Additionally, the parameter “$r$” refers to the perturbation radius in the $\ell_\infty$-norm ball.

---

> ### Author Response · Authors · 2024-11-20
> **Part 2**
>
> > Q2: After reading the entire paper, I found it challenging to identify the specific question the author aims to address and the associated motivations. While the introduction attempts to outline these points, it is difficult to discern the relationship between the motivation and the problem being addressed. Additionally, there appears to be a disconnect between the problem definition in the introduction and the methods presented. Here are some specific suggestions for clarification:
>
> **Response:** We appreciate the reviewer’s feedback regarding the clarity of our research question and motivation. We have addressed the reviewer's specific concerns in Q2 as follows. In this response, we would like to first clarify the positioning and structure of our paper as a big picture to understand our work:
> - **Core Research Question**: Our work focuses on red-teaming existing protective perturbation methods to develop more effective, efficient, and faithful approaches. We observed that existing purification studies are either inefficient (e.g., IMPRESS requires heavy iterative optimization) or produce unfaithful results (e.g., GrIDPure yields hallucinated images). Additionally, both methods operate solely on the input side, limiting the comprehensiveness of red-teaming. Our core motivation is to bridge this gap by proposing a more comprehensive and effective red-teaming framework.
> - **Core Contributions**: Unlike previous works that primarily conduct empirical red-teaming, we are the first to introduce the perspectives of shortcut learning and causal analysis to understand the underlying mechanisms of how protective perturbations affect personalized diffusion model fine-tuning. This new understanding allows us to design a systematic red-teaming framework grounded in causal intervention. Without this deep insight into the shortcut learning induced by protective perturbations, designing a systematic and robust red-teaming framework would be challenging.
> - **Connection Between Introduction and Methodology**: We respectfully argue that these two parts actually are tightly connected in our paper. In the introduction, we first point out the current lack of understanding on "why protective perturbations work", which is a more fundamental problem for both protection and red-teaming sides. Then we introduce our main research question, "how to red-team protective perturbations more effectively, efficiently and faithfully", aiming to bridge the gap of existing purification methods. Correspondingly, in the methodology section, we first provide an explanatory framework based on causal and shortcut learning, addressing the “why” question. Building on this insight, we propose a systematic red-teaming framework grounded in causal intervention. Through experiments, we demonstrate the effectiveness, efficiency, and faithfulness of our framework, thus answering the “how” question of our research.

---

> ### Author Response · Authors · 2024-11-20
> **Part 3**
>
> > Q2.1. The introduction states, “The model trained on perturbed data will generate images that are poor in quality, and thus, unauthorized fine-tuning fails.” Does this imply that generating low-quality images of private content protects copyright and privacy? If so, why does the proposed method focus on enhancing image clarity for private content while defining it as a defense
>
> **Response:** We acknowledge the confusion and appreciate the opportunity to clarify. Our work is a red-teaming effort aimed at defeating existing protective perturbations. While protective perturbations degrade image quality to prevent unauthorized fine-tuning, our goal is to overcome these protections and restore high-quality image generation. This is aligned with previous works like IMPRESS and GrIDPure. The value of our red-teaming work lies in:
>
> - **Evaluating Robustness**: By developing more principled and systematic methods to break protective perturbations, we assess their robustness and reveal potential vulnerabilities. This helps prevent a false sense of security among portrait owners and artists who might overly rely on these protections.
> - **Guiding Future Protections**: Our findings can inform the development of more effective and robust protective perturbation methods, enhancing privacy and copyright protections in the future.
>
> > Q2.2. The author mentions that shortcuts are key to avoiding the generation of private personal images. Given this, why does the method seem to eliminate these shortcuts?
>
> **Response:** We understand this confusion is similar to Q2.1 and would like to clarify. In our work, we identify that protective perturbations cause the model to learn shortcut connections between the added noise and the personalized identifier — an unintended association. Our method aims to eliminate these shortcuts to prevent the model from being misled by the perturbations, thereby allowing it to correctly learn the association between the identifier and the original clean images. This elimination is essential for defeating the protective perturbations and restoring high-quality image generation.
>
>
>
> > Q2.3. On line 46, adversarial perturbations are suggested as a means to protect users’ images from unauthorized personalized synthesis. However, line 100 describes an intention to "defend against" this. Could you clarify?
>
> **Response:** Thank you for pointing out this inconsistency. We followed the terminology used in GrIDPure (Zhao et al., 2024) [1], where “defense” refers to methods that defeat protective perturbations. **However, to avoid confusion, we have revised the manuscript to use the term “red-teaming” instead of “defense,” clearly indicating that our work focuses on breaking existing protective perturbations rather than safeguarding them.**
>
> > Q2.4. Additionally, the highlighted question in the introduction, “How to design an effective, efficient, and faithful purification approach is still an open question,” lacks context. Although there is a mention of “Moreover, purification studies are also purposed to further break those protections” in the following sentence, there are no subsequent explanations, particularly concerning how this question connects with the paragraph's earlier discussion.
>
> **Response:** Thank you for highlighting this issue. In the revised manuscript, we have added more context and explanations regarding purification studies, along with appropriate references. We have also revised the introduction in revision to better present the limitations of existing purification methods including the two important baseline methods, IMPRESS and GrIDPure. This provides a smoother transition to our main motivation of developing a more effective, efficient, and faithful purification approach.
>
> > Q2.5. In the end of introduction, it seems that the authors propose a new purify methods, "Our approach conducts comprehensive purification from three perspectives, including input image purification, contrastive decoupling learning with the negative token, and quality-enhanced sampling....". However, in the methods, the author says they propose a method to address the short cut learning...., which is a little bit confusing.
>
> **Response:** Our method is indeed a purification approach, but it is theoretically grounded in causal analysis and specifically designed to address shortcut learning. Through our causal analysis, we discovered that protective perturbations create shortcut connections during the fine-tuning process, causing the model to learn superficial patterns rather than meaningful semantic features. By understanding and targeting this root cause, our method provides a systematic and principled purification mechanism against protective perturbations. This theoretical grounding distinguishes our approach from previous methods that focus solely on noise removal without considering the underlying causal mechanisms.

---

> ### Author Response · Authors · 2024-11-20
> **Part 4**
>
> > Q3. Minor: Although viewing fine-tuning from a causal effect and shortcut learning perspective is novel, it shares similarities with backdoor attacks. In the backdoor attack literature, several papers have employed causal graphs to analyze shortcut mechanisms.[1-3]
>
> **Response:** Thank you for pointing out these related works in backdoor attacks and defenses. We have added a dedicated subsection in the revised manuscript (Appendix E.1) to discuss these causality-based backdoor defense works. While there are similarities in problem formulation and technical direction, our work is fundamentally different from these studies. Specifically:
>
> - **Different Problem Settings**: Backdoor defenses typically focus on classification tasks, where the spurious correlations are between input features and class labels. In contrast, our work addresses personalized generation tasks in diffusion models, dealing with associations between textual identifiers and image content.
> - **Distinct Technical Approaches**: Our method introduces a unified framework employing both do-calculus (for purification) and decoupling learning, operating on prompt augmentation and image generation, which differs from the techniques used in the backdoor defense literature.
>
> We provide detailed discussions comparing our work with CBD (Zhang et al., 2023) [1] and FABE (Zheng et al., 2024) [2] in the Appendix E.1. We also attached them here for your reference.
>
> > **Comparison with CBD and FABE.** Our work and these works both leverage causality-based perspectives to defend or red-teaming the perturbation. However, the problem and techniques in our work are fundamentally different from these two works. First, in terms of problem, CBD and FABE both focus on the classification task, either image classification or text classification, where the backdoor spurious path is established between the model input $X$ and class label prediction $Y$. For our task, we are tackling the personalized generation task, where the LDMs are fine-tuned to link a unique identifier $\mathcal{V}^*$ to a new subject concept $X_0$. In the backdoor attack case, the attacker aims to introduce a confounder $A$ variable at the input side to trigger certain label prediction $Y'$, while in our case, the image protector only modifies the learning target $X_0^\prime=X_0+\Delta$ but do not explicitly add any trigger at the input side, which serves as the confounder in backdoor attack case. Thus, considering the difference in the threat model, the defense techniques in backdoor case, such as CBD and FABE, focus more on removing the confounder in the input side, while the defense in our case focuses on the prediction side, by reinforcing the causal path between the unique identifier $\mathcal{V}^*$ and the clean target concept $X_0$.
>
> > Second, in terms of techniques, both CBD and FABE only focus on one perspective on causal intervention, while our work proposes a unified framework that conducts both do-calculus (i.e., removing the injected variable or purification) and decoupling learning. Specifically, CBD assumes that the correlations $A \rightarrow Y$ can be well captured by an early-stop model $f_B$, and CBD learns the clean model $f_C: X \rightarrow Y$ by minimizing the mutual information between the embedding from $f_B$ and $f_C$. Compared to this feature space decoupling learning, our work operates the prompt augmentation side, which can be more efficient and end-to-end. Specifically, we observe the fact that the class-specific image doesn't contain any perturbation, while the instance image might contain the perturbation. Thus, we introduce a new noise identifier $\mathcal{V}^*_N$ and append it to two different datasets with different prefixes "with" and "without" to achieve contrastive decoupling learning without any need to access the model weights and tuning any early stopping hyper-parameters as in CBD.
>
> > Similar to the purification part in our work, FABE mainly focuses on conducting semantic denoising on the original textual input to approximately achieve the do-calculus from the causal intervention perspective. Specifically, FABE denoise the $X$ to semantically equivalent text $Z$, with a fine-tuned language model. The fine-tuned language model learns to rank that effective $ Z$ that removes confounder $A$, i.e., the backdoor trigger. Then, the prediction is conducted via voting over a pool of sampled $Z$ to achieve a clean prediction of $Y$. Compared to FABE, our purification pipeline for protective perturbation is more direct and flexible, without the need to fine-tune an additional model. Meanwhile, FABE requires unrolling $B$ semantic candidates using beam search, which can be computationally expensive especially when context length $L$ is large. In contrast, we leverage off-the-shelf image restoration and super-resolution models to conduct one-shot efficient purification.

---

> ### Author Response · Authors · 2024-11-20
> **Part 5**
>
> > Q4. The causal graph is underexplained and possibly contains ambiguities. For example, the definitions of $\bar{c}$ and $\bar{x}_0$ are missing. While a brief introduction to the construction of the graph is provided, explanations of each node’s meaning and the meaning of the arrows are absent. Given that the causal graph is a key contribution, adding a paragraph to introduce and explain it in detail would be beneficial. The term "spurious path" may also be misapplied; in causal inference, this usually refers to a backdoor path between treatment and outcome. Since this doesn’t apply here, either avoid the term or define it within the paper's context.
>
> **Response:** We appreciate the feedback and agree that the causal graph needed a more detailed explanation. While we did provide some definitions of variables in the preliminary section, we acknowledge that these variables' definitions could be more explicit and better integrated into the causal graph discussion part. In the revised manuscript, we have added a dedicated paragraph (Appendix C.1) that thoroughly explains each component of the causal graph, including clear definitions of variables like $\bar{c}$ and $\bar{X}_0$, as well as the meanings of the nodes and arrows. Additionally, to avoid confusion, we have replaced the term “spurious path” with “identifier-noise shortcut” within our context. We also attached the context of causal graph details (in Appendix C.1) here for your reference.
>
> > Appendix C.1. Detailed Explanation of the Causal Graph Building Process
> >
> > To understand how protective perturbations lead to shortcut learning in PDMs, we construct a Structural Causal Model (SCM) that captures the learned causal relationships between the variables involved in the fine-tuning process. The variables in our SCM are defined as follows: $X_0$ represents the original clean images representing the true concept; $\Delta$ denotes the protective perturbations added to the images; $X_0^\prime = X_0 + \Delta$ are the perturbed images used for fine-tuning; $c$ represents class-specific textual prompts without the unique identifier (e.g., "a photo of a person"); $\mathcal{V}^*$ is the unique identifier token used in personalized prompts (e.g., "sks"); $c^{\mathcal{V}^*} = c \oplus \mathcal{V}^*$ denotes the personalized textual prompts combining $c$ and $\mathcal{V}^*$; $\theta_{T}$ represents the model parameters after being fine-tuned. The structural equations governing the relationships in our SCM are as follows: (1) Perturbed Images: $X_0' = X_0 + \Delta$, where $X_0'$ represents the perturbed images, $X_0$ the original clean images, and $\Delta$ the protective perturbations. (2) Model Fine-tuning: $\theta_{T} = f_{\theta}(\theta_0, X_0', c^{\mathcal{V}^*}, \bar{X_0} ,\bar{c})$, where $\theta_{T}$ represents the fine-tuned model parameters, $\theta_0$ the initial model parameters, $c^{\mathcal{V}^*}$ the personalized text prompts, $\bar{X_0}$ and $\bar{c}$ the image and prompt of class-specific dataset to help model maintain class prior. For our case of finetuning on human portrait, the $\bar{X_0}$ is the person images from different identities, and $\bar{c}$ is set as "a photo of person". After $\theta_T$ has been fine-tuned, it learns the latent causal relationship $\mathcal{V}^*$ $\rightarrow$ $X_0'$ with conditioning mechanism through prompt-image association.
>
> ---
>
> > Q5. The causal graph may need structural revision. In causal inference, an arrow between A and B signifies that A causes B. However, in this graph, it seems that an arrow signifies containment rather than causation. I would suggest adhering closely to causal inference conventions and adjusting the graph accordingly.
>
> **Response:** Thank you for this suggestion. We have revised the causal graph in the manuscript to adhere more closely to causal inference conventions, ensuring that the arrows correctly represent causal relationships. In Appendix C.1, we have also provided detailed explanations of each node and edge in the graph to clarify their meanings. We also attached them here for your reference.
>
> > Appendix C.1. Detailed Explanation of the Causal Graph on Node and Edge
> >
> > In the graph, we define each node to represent one of the elements for the learned causation: independent variables (i.e., text prompts, and unique identifier), dependent variables (i.e., perturbed identity images, general face images), or intermediate variables like prompt composited. We define each edge to represent the causal unidirectional dependency between the variables. For those prompt composition edges, the relationship is simply the concatenation operation in the textual space. For those prompt-image association edges, the relationship is defined as the causation learned by the model $\theta_T$. For the edges between $\Delta$ and $X_0'$, it is defined as the direct effect of the perturbations on the original clean images, $X_0'= X_0 + \Delta$.

---

> ### Author Response · Authors · 2024-11-20
> **References**
>
> Again, we appreciate the reviewer for their valuable comments, which helped improve the manuscript significantly.
>
> **References**
>
> `[1]`. Zhao, Zhengyue, et al. "Can Protective Perturbation Safeguard Personal Data from Being Exploited by Stable Diffusion?." Proceedings of the IEEE/CVF Conference on Computer Vision and Pattern Recognition. 2024.
>
> `[2]`. Zhang, Zaixi, et al. "Backdoor defense via deconfounded representation learning." Proceedings of the IEEE/CVF Conference on Computer Vision and Pattern Recognition. 2023.
>
> `[3]`. Liu, Yiran, et al. "Causality Based Front-door Defense Against Backdoor Attack on Language Models." Forty-first International Conference on Machine Learning. 2024.

---

> > ### Comment · Reviewer_uRZe · 2024-11-23
> > **Respond to the rebuttal**
> >
> > Thank you for the authors' reply. After taking a closer look at the paper, I found the overall pipeline clearer, and I now understand the motivation and methods presented in the paper.
> >
> > However, I still have some concerns regarding the novelty of the work. Although the authors provided a causal analysis, it appears to be limited to constructing a causal graph with prior knowledge to describe the problem (typically, this analysis process is highly similar to the causal analysis in backdoor attack, which has been already proposed), without offering theoretical guarantees. Moreover, the proposed methods are not fundamentally based on the causal aspect itself. Upon closer examination, the methods share significant similarities with prior approaches that use purification techniques.
> >
> > The main difference between the previous models and the current one seems to be that the current approach leverages off-the-shelf pretrained methods, whereas the previous models are optimization-based. Additionally, the only key modification involves changing the image prompts, such as appending a suffix like "with/without XX noisy pattern" to the images.
> >
> > For these reasons, I have decided to maintain my score.

---

> > > ### Author Response · Authors · 2024-11-24
> > >
> > > Thank you for taking the time to reply to our response. According to the 2021-2025 ICLR Reviewer Guide, novelty should be evaluated based on both **technical methods** and **novel findings**. We believe our work contributes significantly in both aspects. Below, we provide detailed clarifications regarding the novelty of our work.
> > >
> > > **Contribution Statements:**
> > >
> > > 1. **Novel Finding on Protective Perturbations**: Our work is the first to analyze and rethink the effectiveness of protective perturbations through a causal and shortcut learning lens. We make the novel discovery that **effective protective perturbations create latent-space image-prompt mismatches**. This means that the perturbed images and their corresponding prompts are no longer semantically aligned in the latent space. We validate this finding through extensive experiments, including latent mismatch visualizations and concept interpretations (Section 4.1 and Appendix B.2).
> > >
> > > 2. **Systematic Red-Teaming Framework**: Building on this insight, we propose a systematic, efficient, and faithful red-teaming framework against existing protective perturbations. Our framework achieves state-of-the-art performance across 9 purification baselines and 7 protection methods, excelling in defense effectiveness, efficiency, and faithfulness. This extends beyond existing purification techniques that solely focus on image denoise, incorporating decoupling learning to address the limitations of prior methods.
> > >
> > > 3. **Novel Contrastive Decoupling Learning (CDL)**: Our CDL method is the first to explicitly guide models to separately learn clean and noisy concepts during fine-tuning for personalized generation tasks. Through classifier-free guidance (Eq. 6, Section 4.2) during sampling, our CDL effectively decouples these concepts, enabling robust red-teaming against protective perturbations. We demonstrate that CDL is not only effective on its own but also works synergistically with purification techniques. Moreover, our experiments in Section 5.3 (Resilience Against Adaptive Perturbations) highlight CDL as a robust, and potentially once-for-all solution for breaking protective perturbations.
> > >
> > > These contributions offer significant insights and advancements in both protective perturbation design and red-teaming methodologies, with broader implications for the ICLR community.

---

> > > ### Author Response · Authors · 2024-11-24
> > >
> > > We further provide point-by-point responses to the reviewer's concerns:
> > >
> > > > **Causal Analysis Novelty**: “Although the authors provided a causal analysis, it appears to be limited to constructing a causal graph with prior knowledge to describe the problem, without offering theoretical guarantees.”
> > >
> > > **Response**: While we do not claim to propose a new causal analysis framework or theoretical guarantees, our contribution lies in **being the first to apply causal analysis to protective perturbations in personalized generation tasks**. Through this lens, we made the novel discovery of **latent-space image-prompt mismatches**, validated by extensive empirical evidence (Section 4.1, Appendix B.2).
> > >
> > > Regarding similarities with causal analyses in backdoor attacks: while both involve causal frameworks, the problem settings differ fundamentally. Backdoor analyses typically focus on simple classification tasks (input-output pairs), whereas our work addresses text-to-image diffusion models with complex conditioning and multiple learning targets. Additionally, the confounder in backdoor attacks is introduced at the input level, while in our case, it arises from protective perturbations affecting learning targets. To our knowledge, this is the first causal analysis tailored to personalized generation tasks under adversarial setups. Additionally, this analysis motivated our CDL module, which introduces a noise-related node $\mathcal{V}_N^*$ to decouple clean and noisy concepts—a design not present in prior work.
> > >
> > > > **Purification Pipeline Novelty**: “The proposed methods are not fundamentally based on the causal aspect itself. Upon closer examination, the methods share significant similarities with prior approaches using purification techniques.”
> > >
> > > **Response**: While all red-teaming methods share the common goal of mitigating perturbations, our work differs in both **approach** and **systematic integration**:
> > >
> > > 1. **Beyond Image Denoising**: Unlike prior methods (e.g., GrIDPure, IMPRESS) that focus solely on image-space denoising, our framework incorporates causal insights to address the root cause of latent mismatches. This allows us to go beyond traditional purification, introducing decoupling learning to systematically restore alignment and generation quality.
> > >
> > > 2. **Contrastive Decoupling Learning (CDL)**: Our CDL module not only adds suffixes to prompts but also adjusts the sampling process to enhance decoupling and generation quality. By learning clean and noisy concepts separately, and guiding the model with enhanced classifier-free guidance (Eq. 6), we ensure high fidelity and clarity in generated images.
> > >
> > > 3. **More Efficient, Faithful and Practical Purification**: Leveraging off-the-shelf image restoration and super-resolution models, our approach avoids heavy iterative optimization (comparable to IMPRESS) and produces faithful content (comparable to GrIDPure). Furthermore, leveraging our image restoration models as purification pipelines is also more aligned with real-world standardized practices [1,2] on training personalized diffusion models on potentially corrupted data, providing a more practical red-teaming framework for the protection side.
> > >
> > > We hope these clarifications address the reviewer’s concerns and highlight the novelty and significance of our work. We are happy to discuss any specific points or further elaborate on areas of interest.
> > >
> > > **References**
> > >
> > > `[1]`. Kohya-ss. SD Scripts. GitHub, https://github.com/kohya-ss/sd-scripts. Accessed 23 Nov. 2024
> > >
> > > `[2]`. Akegarasu. LoRA & Dreambooth Training Scripts & GUI. GitHub, https://github.com/Akegarasu/lora-scripts. Accessed 23 Nov. 2024.

---

### Official Review · Reviewer_2TWg · 2024-10-31

**Soundness:** 3
**Presentation:** 3
**Contribution:** 3
**Rating:** 6
**Confidence:** 3

**Summary:**

The paper conduct a comprehensive analysis to show that perturbations induce a latent-space misalignment between images and their text prompts in the CLIP embedding space, which leads to association between the noise patterns and the identifiers. Based on this observation, the paper introduces contrastive decoupling learning with noise tokens to decouple the learning of personalized concepts from spurious noise patterns.

**Strengths:**

1. The observation that adversarial perturbations induce a latent-space misalignment between images and their text prompts in the CLIP embedding space is interesting and insightful.

2. The paper is well-organized and easy-to-follow.

3. The paper conducts an extensive array of experiments and also considers adaptive perturbation.

**Weaknesses:**

1. The paper does not provide strong theoretical analysis to support the conclusions.

2. The technical contribution is a little limited since Decoupled Contrastive Learning is not a new technique proposed by the paper.

**Questions:**

I am wondering if the noisy images generated by DM without any defense can be denoised?

---

> ### Author Response · Authors · 2024-11-24
>
> Dear Reviewer 2TWg, we would like to check if our rebuttal has addressed your concerns or if any points still require clarification before the response period ends. Thank you for your time.

---

### Official Review · Reviewer_co1c · 2024-11-02

**Soundness:** 3
**Presentation:** 3
**Contribution:** 3
**Rating:** 8
**Confidence:** 3

**Summary:**

This paper uncovers and validates the underlying mechanism by which adversarial perturbations disturb the fine-tuning of personalized diffusion models by latent-space image-text misalignment. Then, it introduces a systematic defense framework that mitigates the misalignment with data purification and contrastive decoupled learning and sampling.

**Strengths:**

- This paper finds that adversarial perturbation leads to latent image-text mismatch and provides an explanation from the perspective of shortcut learning. Their analysis contributes to the further development of protective perturbation in personalized diffusion models.

- The proposed framework provides a system-level defense covering data purification, model training, and sampling strategy. Compared with previous data transformation and diffusion-based methods, the proposed method achieves the best semantic and image quality restoration.

**Weaknesses:**

- In Table I, the authors would better add a setting that the clean images are processed by the proposed and baseline methods.

- In Table II, why only calculate the time for data purification? Will CDL incur additional time costs?

**Questions:**

Please help to check weaknesses.

**Details Of Ethics Concerns:**

This is a promising method to destroy nearly all SOTA defense study on personalized diffusion model.  As it mentioned, it provides a valuable evaluation framework.  However, how to protect is still unsolved.

---

> ### Comment · Reviewer_co1c · 2024-11-26
>
> I acknowledge that I have read the response.

---

### Official Review · Reviewer_nipZ · 2024-11-04

**Soundness:** 2
**Presentation:** 1
**Contribution:** 2
**Rating:** 6
**Confidence:** 3

**Summary:**

The paper aims to improve the personalization performance of Diffusion Models on images with protective perturbation, a kind of noise avoiding images to be learned by models. The authors fist empirically analyze the latent mismatch between the perturbed and original images, finding that perturbation significantly alternate the latent representations of images. The authors believe that the mismatch causes shortcut learning and therefore fail the personalization of diffusion models on such perturbed data. Therefore, a novel method is proposed to improve the personalization training by contrastive learning and super resolution.

**Strengths:**

1. The proposed contrastive learning method is well motivated by the empirical finding on the latent mismatch of perturbed images.
2. In multiple domains, the method presents better fine-tuning performance than baselines given protective perturbation on images.
3. Comprehensive experiments are conducted to understand and evaluate the method.

**Weaknesses:**

1. It is not clear to me the connection between the latent mismatch and the shortcut learning. Why does the existence of latent mismatch lead to shortcut learning?
2. I don't think the word "defending" (in the title) should be used against a good technique, protective perturbation. The paper is a good red-teaming paper that explored a stronger threat model for protective perturbation. Unfortunately, many description of the method is defined as a mitigation method, which could mislead the readers about the negative impacts of the methods. The authors should discuss how this method can break the existing protective perturbation. It would be appreciated if the authors can discuss potential solutions toward better copyright protection via protective perturbation or other alternatives.

**Questions:**

* It is not clear to me the connection between the latent mismatch and the shortcut learning. Why does the existence of latent mismatch lead to shortcut learning?
* What are the potential mitigation against to the proposed method?

**Details Of Ethics Concerns:**

The proposed method can put the copyright of artists' work at risk. The method can void the protective perturbation in protecting images from being used for training diffusion models. The authors did not discuss the potential negative impacts.

---

> ### Author Response · Authors · 2024-11-24
>
> Dear Reviewer nipZ, we would like to check if our rebuttal has addressed your concerns or if any points still require clarification before the response period ends. Thank you for your time.

---

### Author Response · Authors · 2024-11-20
**General Response**

We greatly appreciate all reviewers for your time and effort in providing this insightful feedback that helps us improve our work. We have submitted a revised version of the paper that highlights the changes in blue color. In this post, we provide a general response summary to the most common questions and the main updates in our revision.

### Response Summary to Common Questions

> Q1: Analysis and connection between latent mismatch and shortcut learning (Reviewer `nipZ`, `2TWg`)

**Response**: We provide a more detailed explanation of how latent mismatch leads to shortcut learning. In Section 4.1, we demonstrate that protective perturbation transforms the target to $X_0^\prime = X_0 + \Delta$, causing semantic displacement in the latent space toward noise patterns rather than preserving the original identity concept. Given the architectural constraints of Dreambooth learning, where $\mathcal{V}^*$ must associate with either noise or identity concept, the model naturally converges to learning spurious correlations between $\mathcal{V}^*$ and $\Delta$ as this represents the path of least resistance for loss minimization.

---

> Q2: Positioning and framing of the work as red-teaming protective perturbations rather than safeguarding them (Reviewer `uRZe`, `nipZ`)

**Response**: We appreciate the feedback about the positioning of our work. We have made several important revisions:

- Updated the "Defending" in title to "Red-Teaming" to better reflect our positioning to avoid confusion.
- Clarified throughout the paper that our work is a red-teaming effort aimed at understanding and further breaking protective perturbations.
- Updated the introduction to focus on the two key related works, including IMPRESS and GrIDPure, clarifying terminology, and maintaining consistent terminology around "protective perturbations".

---

> Q3: Technical novelty of contrastive decoupling learning and comparison with existing work in backdoor defense (Reviewer `uRZe`, `2TWg`)

**Response**: While decoupling learning exists in backdoor defense literature [1] as pointed out by reviewer `uRZe`, our approach differs substantially in both problem space and technical implementation from CBD [1]. We believe that our work makes a novelty contribution in terms of methodology and also findings. We summarize the main difference with CBD [1] here for reviewers' convenience:
1. We address fundamentally different challenges than previous work like CBD [1]. CBD focuses on backdoor attacks in classification tasks, where attackers introduce input-side confounders to trigger specific label predictions, whereas our work tackles personalized generation in text-to-image diffusion models where protectors modify the learning target ($X_0^\prime=X_0+\Delta$) to link identifiers ($\mathcal{V}^*$) with subject concepts.
2. Our technical approach achieves decoupling more efficiently through prompt augmentation that capitalizes on inherent differences between class-specific and instance-specific images. In CBD, decoupling is achieved through feature space decoupling through mutual information minimization between early-stop and clean models. However, our approach achieves more efficient decoupling through prompt augmentation by leveraging the inherent differences between class-specific and instance-specific images. We introduce a noise identifier $\mathcal{V}^*_N$ and use "with"/"without" prefixes for contrastive decoupling learning, eliminating the need for model weight access or complex early-stopping parameters that CBD requires.

Please refer to Appendix E.1 in the revised manuscript for more discussion with existing work in backdoor defense.

---

### Change Summary in the Revised Version
1. Improved introduction by focusing on the two key related works, including IMPRESS and GrIDPure, clarifying variables like $r, \bar{c},\bar{X_0}$, and maintaining consistent terminology around "protective perturbations". (`uRZe`)
2. Updated the causal graph in Figure 2 and Figure 9 with more detailed construction details in Appendix C following the conventional notation in causal inference. (`uRZe`)
3. Enhanced clarity throughout the paper regarding our positioning as a red-teaming effort. (`uRZe, nipZ`)
4. Expanded discussion in Section 4.1 to better explain the connection between latent mismatch and shortcut learning. (`uRZe, nipZ`)
5. Added detailed comparison with related work in backdoor defense in Appendix E.1. (`uRZe, 2TWg`)
6. Added comprehensive discussion of alternative copyright protection approaches in Appendix E.2. (`nipZ`)
7. Added results under the clean setup in Table 1 to evaluate purification methods' performance without protective perturbations. (`co1c`)
8. Added post-hoc purification results on noisy outputs in Appendix B.6 and Figure 8. (`2TWg`)
9. Added discussion on broader impact and more adaptive protection in Appendix B.5. (`nipZ, co1c`)

**References:**

`[1]`. "Backdoor defense via deconfounded representation learning." CVPR'23

---

> ### Author Response · Authors · 2024-11-26
>
> We additionally provide a point-by-point response to the reviewer's ethics concerns:
>
>
> > `nipZ`: The proposed method can put the copyright of artists' work at risk. The method can void the protective perturbation in protecting images from being used for training diffusion models. The authors did not discuss the potential negative impacts.
>
> **Response**: **We additionally discuss the potential negative impacts of our work in the broader impact sector in Appendix B.5 in the revised manuscript.** We attached them here for your reference.
>
> > **Discussion on Broader Impact:** Our work on red-teaming existing protective perturbations raises ethical considerations, particularly regarding privacy and intellectual property rights. While our methods could potentially compromise images protected by existing protective perturbations, we believe that the benefits of this research outweigh the potential risks. First, our research helps prevent a false sense of security by revealing limitations in existing protective measures. This transparency enables portrait owners and artists to make more informed decisions about protecting their content. Furthermore, the insights gained from our analysis can inform the development of next-generation protection techniques that are more resilient against sophisticated red-teaming, thereby strengthening privacy and copyright safeguards in the long term.
>
> ---
>
> > `co1c`: This is a promising method to destroy nearly all SOTA defense study on personalized diffusion model. As it mentioned, it provides a valuable evaluation framework. However, how to protect is still unsolved.
>
> **Response**: Our work primarily focuses on red-teaming and provides a comprehensive evaluation framework for existing protective perturbations. While we demonstrate that our framework is robust against adaptive perturbations (Section 5.3), we acknowledge that more sophisticated protection techniques may emerge. For instance, our red-teaming setup currently focuses on noise-based protective perturbations, but object-embedded perturbations (Zhu et al., 2024) could potentially resist our noise-concept-based CDL prompt design. Additionally, to counter our purification pipeline, future protection techniques could explore more advanced ensemble methods (Chen et al., 2022) to develop more resilient defenses. **We have added this discussion to the limitations section in Appendix B.5 of the revised manuscript.**

---

### Comment · Area_Chair_WRmW · 2024-11-24

Dear reviewers,

Thanks for serving as a reviewer. As the discussion period comes to a close and the authors have submitted their rebuttals (maybe in general response), I kindly ask you to take a moment to review them and provide any final comments.

If you have already updated your comments, please disregard this message.

Thank you once again for your dedication to the OpenReview process.

For authors, I think it will be better to also respond to each reviewer separately, as it will be easier for reviewers to find whether their own concerns have been addressed.

Best,

Area Chair

---

> ### Comment · Reviewer_co1c · 2024-11-26
>
> I personally quite like this paper in the reviewing phrase.  However, after reading other comments,  I find out there are some room for it to update and improve in the revision.  I would not like to champion for its acceptance.
>
> Only one point for Reviewer uRZe, to set up a causal graph without any guarantee is a common way in that domain (I do not support it either).  Thus, I think you two might be not one the same page in the discussion :)

---

> > ### Author Response · Authors · 2024-11-26
> >
> > Thank you for your input regarding the `uRZe`’s concern on the theoretical guarantee. Aligned with previous works in the backdoor domain (e.g., Zhang et al., 2023; Liu et al., 2024), we use the causal graph primarily as a conceptual tool to illustrate the learning relationships and to motivate our methodology. Our intention is to provide intuitive insights into the mechanism of protective perturbations that can guide the development of our red-teaming framework.

---

> ### Comment · Reviewer_uRZe · 2024-11-28
> **Concerns from Reviewer uRZe**
>
> Thank you author for your responds. In your response “our contribution lies in being the first to apply causal analysis to protective perturbations in personalized generation tasks” And based the current version of the causal part in your paper, I would not think it is a contribution or should be called as causal analysis.
>
> Typically, I agree that most causal graphs are constructed using prior knowledge, drawing from experience in a specific domain to determine which features cause others, and representing these relationships in a graph [3-4]. These graphs serve as **assumptions** for further causal treatment estimation or other in-depth causal analyses, which I believe are not present in this paper. The papers cited by the authors that incorporate causal analysis into their work not only construct causal graphs but also provide in-depth causal analyses (e.g., using front-door adjustment [2]) or design models explicitly based on causal theory (e.g., disentangling causal and confounding factors [1]). Hence, I do not consider merely constructing a causal graph to be an analysis of the problem. Instead, it appears to be another way of **describing the problem** from the author’s perspective, which is also pointed out by reviewer co1c.
>
>
> Additionally, regarding the method described in the paper: While I acknowledge the effectiveness of the approach, I believe that, compared to previous methods, the primary difference lies in the addition of a suffix prompt. This distinction, however, may not constitute a particularly strong novelty. Also the method in the paper "During training, we insert V ∗ N into the prompt of instance data with the suffix “with XX noisy pattern”, and include the “inverse” of V ∗ N in the prompt of class-prior data with the suffix “without XX noisy pattern”. During inference, we add the suffix “without XX noisy pattern” to the prompt input to guide the model in disregarding the learned patterns associated with V ∗ N. It seems, in essence, that this method functions like injecting a backdoor attack into the model. Specifically, all instance data is linked to the suffix “with XX noisy pattern,” while class-prior data is linked to “without XX noisy pattern.” As a result, during inference, if the prompt includes the suffix “without XX noisy pattern,” the model generates a safe image. However, this mechanism resembles that of a backdoor attack, where the attacker could potentially exploit it. For example, if the attacker becomes aware of this mechanism and inputs the suffix “with XX noisy pattern,” I suspect all personalized images would be generated, compromising the intended safety.
>
> Due to these concerns, I maintain my opinion on the paper. That said, I am happy to engage in further discussion with other reviewers to hear their perspectives and clarify any points.
>
>
> [1]Zhang Z, Liu Q, Wang Z, et al. Backdoor defense via deconfounded representation learning[C]//Proceedings of the IEEE/CVF Conference on Computer Vision and Pattern Recognition. 2023: 12228-12238.
>
> [2]Liu Y, Xu X, Hou Z, et al. Causality Based Front-door Defense Against Backdoor Attack on Language Models[C]//Forty-first International Conference on Machine Learning.
>
> [3]Pearl J. Causal inference in statistics: An overview[J]. 2009.
>
> [4] Yao L, Chu Z, Li S, et al. A survey on causal inference[J]. ACM Transactions on Knowledge Discovery from Data (TKDD), 2021, 15(5): 1-46.

---

> ### Author Response · Authors · 2024-11-29
> **Response to Reviewer uRZe’s Comments and Concerns**
>
> Thank you for your detailed feedback. We would like to address the concerns you mentioned, and we hope this response provides clarity and facilitates further discussion among reviewers.
>
> > Q1. Contribution and Depth in Causal Analysis Part
>
> **Response:** We agree that in-depth causal analysis should not be listed as a key contribution of our work. Instead, our contribution lies in providing an in-depth understanding of the first research question: *Why do existing protective perturbations work?* This led us to uncover the latent-space image-prompt mismatch, which we identify as a key mechanism exploited by existing protective perturbation methods.
>
> Using a causal graph to describe and explain the problem, we identified the **identifier-noise shortcut path** as the root cause of the protective perturbation effect. Further, we demonstrated that this shortcut path does not activate by default with random perturbations, but rather through the latent-space image-prompt mismatch—a novel mechanism we discovered. This hypothesis was validated with extensive experiments, including latent visualizations and interpretation studies (Figures 3, 7, 9; Appendix B.2 in the revised manuscript).
>
> In Section 4.1, we provide an in-depth analysis connecting the effectiveness of perturbations to the latent-space mismatch hypothesis. While we acknowledge that our methodology includes elements of empirical causality-based defense, our work contributes a broader concept that incorporates both **do-calculus** and **decoupling learning strategies**—a conceptual advancement over works like CBD and FABE. More details can be found in Appendices E.1 and C.2 of the revised manuscript. Taken together, our systematic defense strategies and state-of-the-art red-teaming results represent a significant contribution to the field.
>
>
> > Q2. Purification Part Does Not Differ from Previous Works
>
> **Response:** We respectfully disagree with this assessment. Our purification pipeline introduces significant differences compared to prior methods like IMPRESS and GrIDPure, both of which leverage off-the-shelf diffusion models (e.g., Stable Diffusion in IMPRESS and pre-trained unconditioned diffusion models in GrIDPure) as denoisers. In contrast, our work is the first to explore the use of **image restoration models and super-resolution models** for handling adversarial perturbations in protective perturbation tasks. Unlike the iterative optimization approach of IMPRESS and the grid-division strategy of GrIDPure, our pipeline focuses on designing an effective combination of modules, validated through adaptive perturbation experiments and ablation studies. Additionally, our method addresses practical challenges, such as **inefficiency and hallucination issues**, observed in previous purification techniques. This practical contribution should not be overlooked, as it resolves key limitations in existing methods while delivering strong red-teaming results.
>
>
> > Q3. Similarity to Backdoor Attacks and Further Safety Concerns
>
> **Response:** We appreciate this concern and would like to clarify potential misunderstandings about the problem setup and security implications of our method.
>
> Firstly, our work is focused on **red-teaming** protective perturbations to **break the protection effect** and enable the generation of high-quality personalized images from protected datasets. **We are not claiming to generate “safe” images or to enhance the safety of the models in terms of preventing unauthorized use.** Instead, our objective is to counteract protective perturbations crafted to disrupt personalized diffusion model fine-tuning. Specifically, in our problem setup, the image protector crafts protective perturbation that fools the personalized diffusion model fine-tuning process, and our red-teaming side is on retaining the clean generation performance.
>
> Secondly, regarding the potential for further exploitation from the mentioned “attacker” who assumes to have access to the trained models with our method, we respectfully argue that it’s unlikely to happen. Adding the suffix “with XX noisy pattern” during inference would not enable the “attacker” to generate protected personalized images. Instead, it would likely degrade the model’s generation performance because the model learns to associate the “with XX noisy pattern” suffix with the noise patterns introduced by the protective perturbations. **Therefore, there is no incentive for the “attacker” or a model trainer to use this suffix, as it would not yield beneficial results.**
>
> Thirdly, our method fundamentally differs from backdoor attacks in both objective and mechanism (see App E.2). In backdoor attacks, triggers are intentionally injected to create spurious correlations between a trigger and the target label, allowing attackers to manipulate the model. Our approach aims to **decouple spurious correlations**, restoring the correct association between personalized identifiers and clean concepts.

---

### Meta-Review · Area_Chair_WRmW · 2024-12-21

**Metareview:**

The paper proposes  a view for the fine-tuning process of Personalized Diffusion Models (PDMs) as shortcut learning, motivated by causal analysis. The authors introduce a defense framework to help the model correctly associate images with their original semantic meanings.

Strength:
1. The paper studies why protective noise works in T2I models.

2. The paper conducts an extensive array of experiments and also considers adaptive perturbation.

Weaknesses:

The authors and reviewers discuss about the causal analysis in this paper and even the authors agree that their causal analysis is not strict. I think this greatly weaken this paper bacuase of the following reasons, 1. short-cut analysis are not a newly proposed term in unlearnable exmaple, and classification also has "class-image feature misalignment". Due to this reason, I believe the authors need to add some new insights to make this paper stand out, like causal analysis. Therefore, their causal analysis should not be just drawing some causal graphs. 2. In the paper's contribution, the causal analysis also mentioned a lot by the authors. Therefore, I think the authors should do in-depth analysis to demonstrate their analysis are correct.

Therefore, I think this paper still need a lot modifications before the acceptance.

**Additional Comments On Reviewer Discussion:**

The authors and reviewers dicuss a lot about the paper's causal analysis, which should be the key novelty and contribution of this paper.

---

### Decision · Program_Chairs · 2025-01-22

Reject